# Orphan quality control by an SCF ubiquitin ligase directed to pervasive C-degrons

Ka-Yiu Edwin Kong [1,2,3] ✉, Susmitha Shankar [1,2], Frank Rühle [1] & Anton Khmelinskii [1,3] ✉

Selective protein degradation typically involves substrate recognition via short linear motifs known as degrons. Various degrons can be found at protein termini from bacteria to mammals. While N-degrons have been extensively studied, our understanding of C-degrons is still limited. Towards a comprehensive understanding of eukaryotic C-degron pathways, here we perform an unbiased survey of C-degrons in budding yeast. We identify over 5000 potential C-degrons by stability profiling of random peptide libraries and of the yeast C-terminome. Combining machine learning, high-throughput mutagenesis and genetic screens reveals that the SCF ubiquitin ligase targets ~40% of degrons using a single F-box substrate receptor Das1. Although sequence-specific, Das1 is highly promiscuous, recognizing a variety of C-degron motifs. By screening for full-length substrates, we implicate SCF$^{Das1}$ in degradation of orphan protein complex subunits. Altogether, this work highlights the variety of C-degron pathways in eukaryotes and uncovers how an SCF/C-degron pathway of broad specificity contributes to proteostasis.

Selective protein degradation by the ubiquitin-proteasome system (UPS) contributes to proteostasis by removing unnecessary or abnormal proteins, including those arising because of misfolding, failure to assemble into complexes or to reach the correct subcellular compartment[1–5]. Accumulation of abnormal proteins is potentially toxic and has been linked to various neurodegenerative disorders, cancers and aging, highlighting the importance of understanding how proteins are selectively targeted for degradation.

Substrates of selective protein degradation are first recognized and marked with ubiquitin or polyubiquitin chains for subsequent degradation by the proteasome. The selectivity of the recognition process is typically provided by ubiquitin ligases, which recognize specific signals known as degrons in their substrates[6,7]. Degrons are usually short linear motifs found in intrinsically disordered regions, including protein N- and C-termini, and can be recognized based on their unique amino acid sequences or biophysical properties. The first degrons to be discovered are located at protein N-termini[8,9]. These are arguably the simplest N-terminal degrons (N-degrons) whereby only the identity of the N-terminal amino acid determines the stability of a protein. In the classical Arg/N-degron pathway[10,11], previously known as the N-end rule pathway, full-length proteins or protein fragments resulting from endoproteolytic cleavage that start with positively charged (arginine, lysine or histidine) or hydrophobic (leucine, phenylalanine, tryptophan, tyrosine, isoleucine) residues are recognized by the UBR ubiquitin ligases (Ubr1 in yeast[12], UBR1, UBR2 and UBR4 in mammals[13]). Remarkably, the Arg/N-degron pathway is conserved across evolution from bacteria, which lack a bona fide ubiquitin system, to mammals, indicating its fundamental role in protein degradation[10].

Since the discovery of the Arg/N-degron pathway, various ubiquitin ligases have been found to target specific N-degrons. These include the GID complex (CTLH in mammals) that targets proteins via proline or threonine N-degrons[14–18], the Doa10 ubiquitin ligase (MARCH6 in humans) that can recognize substrates via acetylated N-termini[19,20] and IAP ubiquitin ligases, which can target substrates upon failure of N-terminal acetylation[21]. Despite this progress in the understanding of N-degron pathways, studies of C-terminal degrons (C-degrons) have lagged behind.

[1]Institute of Molecular Biology (IMB), Mainz, Germany. [2]These authors contributed equally: Ka-Yiu Edwin Kong, Susmitha Shankar. [3]These authors jointly supervised this work: Ka-Yiu Edwin Kong, Anton Khmelinskii. ✉e-mail: k.kong@imb-mainz.de; a.khmelinskii@imb-mainz.de

Recent studies in human cells have revealed a multitude of C-degrons recognized by cullin-RING ubiquitin ligases (CRLs), the CRL2 and CRL4 complexes in particular[22,23]. Using distinct substrate receptor subunits, CRL2 complexes can recognize several C-degron motifs ending in glycine or arginine, which can differ by as little as a single amino acid. Such motifs, and motifs ending with glycine in particular, are depleted from C-termini of human proteins, presumably to avoid unwanted protein degradation and focus the pathways on specific targets[23]. Interestingly, human C-degron motifs ending with glycine are depleted from the proteomes of yeast and other eukaryotes[23,24], pointing towards potential conservation of C-degrons across eukaryotes. However, in contrast to most targeting factors of the N-degron pathways, which appear to be conserved from yeast to humans[10], the CRL2 complex is not, as the CRL2 scaffold, the cullin Cul2, is absent in yeast[25].

Here, we sought to address this conundrum and assess the conservation of C-degron pathways with budding yeast as a model. Using unbiased in vivo approaches, we find that, despite the absence of CRL2 ubiquitin ligases, sequence-specific C-degrons are pervasive in yeast. However, these degrons do not fall into typical CRL2 or CRL4 motifs but follow distinct rules. Remarkably, most yeast C-degrons are recognized by the conserved SCF/CRL1 ubiquitin ligase using the substrate receptor subunit Das1. Thus, unlike the CRL2 and CRL4/C-degron pathways, which rely on multiple receptors to recognize substrates via distinct C-degrons, the SCF/C-degron pathway identified here appears to employ a single substrate receptor with broad specificity to target a variety of C-termini.

## Results

### Search for C-degrons with libraries of random peptides

We sought to systematically identify C-degrons in the budding yeast *Saccharomyces cerevisiae*. To do so in an unbiased fashion, we decided to use multiplexed protein stability (MPS) profiling[26] to screen libraries of random peptides for C-degrons. In MPS profiling, libraries of peptides, encoded in pools of DNA oligonucleotides, are fused to a tandem fluorescent protein timer (tFT). A tFT consists of two fluorescent proteins that differ in their folding and maturation rates, e.g., mCherry and sfGFP. The mCherry/sfGFP ratio of fluorescence intensities in steady state anti-correlates with the turnover of tFT-tagged proteins and should decrease with increasing turnover of tFT-peptide fusions[27] (Fig. 1a). tFT-peptide libraries can be sorted into eight stability bins according to the mCherry/sfGFP ratio using fluorescence activated cell sorting (FACS), followed by deep DNA sequencing to identify the peptide sequences present in each bin. The distributions of sequencing reads are finally summarized in the form of a protein stability index (PSI) for each peptide, scaled between 0 (unstable) and 1 (stable) (Fig. 1a, Methods).

We constructed tFT-peptide libraries with random peptides 4, 8 or 12 amino acids in length. Using flow cytometry, we assessed the percentage of cells with low mCherry/sfGFP ratios (below those in a control strain expressing only the tFT) as a proxy for the percentage of putative degrons in each library. The flow cytometry profiles suggested that only the library with 12 amino acid long peptides (tFT-$X_{12}$) had a substantial fraction of putative degrons (Fig. 1b). Two factors could contribute to the apparent lack of degrons in the tFT-$X_4$ and tFT-

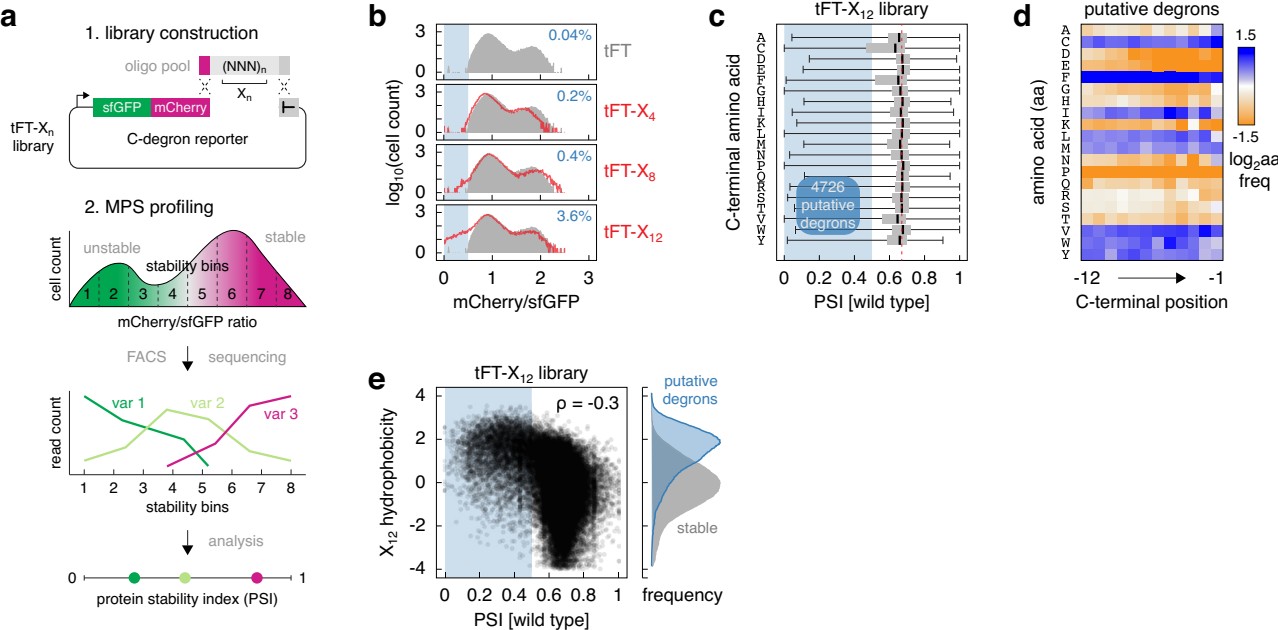

**Fig. 1 | Search for C-degrons using random peptide libraries. a** Workflow of the systematic search for C-degrons. Degenerate oligonucleotides encoding random peptides of defined length (*n*) are cloned into a C-degron reporter plasmid by homologous recombination in yeast. The stability of each sequence variant $X_n$ in the pool, fused to the tandem fluorescent protein timer (tFT), is then determined using multiplexed protein stability (MPS) profiling, whereby the pool is sorted into stability bins using the tFT readout, followed by deep sequencing to determine the frequency of each variant across bins. The distribution of sequencing reads for each variant is summarized by a weighted average of reads across bins in the form of a protein stability index (PSI, Methods) scaled between 0 (unstable) and 1 (stable). **b** Distributions of mCherry/sfGFP ratios, determined by flow cytometry, in pooled yeast libraries expressing random peptides of different lengths (*n* = 4, 8 or 12 amino acids) fused to the sfGFP-mCherry timer (tFT-$X_n$ libraries). A strain expressing the

tFT alone is shown for comparison (grey). Percentage of cells expressing unstable variants (blue region, mCherry/sfGFP <0.5) is shown in each plot. *N* = 35000 cells per sample. **c** Distribution of PSIs in the tFT-$X_{12}$ library determined by MPS profiling. Peptides were grouped by the C-terminal amino acid. Blue region marks putative degrons (PSI < 0.5). Centerlines mark the medians, box limits indicate the 25th and 75th percentiles, and whiskers extend to the minimum and maximum value in each group. **d** Relative amino acid frequency ($\log_2$ transformed) per position in the 4726 putative degrons from the tFT-$X_{12}$ library, normalized to the relative frequency in the whole library. Blue—amino acid enrichment, orange—depletion in the putative degrons. **e** Correlation between peptide hydrophobicity (Kyte-Doolittle hydropathy scale) and PSI in the tFT-$X_{12}$ library. Blue region marks putative degrons. ρ, spearman correlation coefficient. Hydrophobicity distributions for the putative degrons and stable peptides (right). Source data are provided as a Source Data file.

$X_8$ libraries: the relative inaccessibility of C-terminal peptides due to the short linker separating them from mCherry (Methods) and the lack of a long unstructured region that could serve as an initiation site for proteasomal degradation[28,29].

Next, we performed MPS profiling of the tFT-$X_{12}$ library to identify putative degron sequences. The profiling procedure was robust, yielding reproducible PSIs for the peptide sequences detected across replicates (Supplementary Fig. 1a, Supplementary Data 1), and accurate, with a high correlation between PSIs and mCherry/sfGFP ratios of individual constructs measured with flow cytometry (Supplementary Fig. 1b, pearson correlation coefficient of 0.995). Importantly, judging by the amino acid frequencies across the 12 positions in the random peptides, the library composition was uniform (Supplementary Fig. 1c). Hereafter, the 12 amino acids in the $X_{12}$ peptides are numbered from −12 (the most proximal to the tFT) to −1 (the most C-terminal).

Strikingly, over 10% of peptides (4726 out of 46152 detected sequences) were putative degrons, with a PSI below a threshold of 0.5 (Supplementary Fig. 1d, Methods). Grouping all peptides by the C-terminal amino acid showed that a larger fraction of peptides ending in one of three amino acids [CFV] was unstable (Fig. 1c). Moreover, putative degrons were on average enriched in hydrophobic amino acids [CFILMVWY] throughout their sequence, not only at the very C-terminus (Fig. 1d). This suggests that overall hydrophobicity is an important degron determinant in the tFT-$X_{12}$ library. Supporting this notion, peptide hydrophobicity showed a moderate negative correlation with PSI in the library (Fig. 1e). This observation is consistent with prior analyses of terminal peptide libraries both in yeast and human cells[23,26,30–32].

## Identification of C-degron motifs

Despite the prevalence of hydrophobic degrons, we sought to identify sequence-specific degrons and degron motifs in our dataset. Previous efforts to predict degrons have used a variety of approaches, including deep learning[33–36]. Here we decided to combine a deep learning approach with model interpretation using SHapley Additive exPlanations (SHAP)[37]. We trained a deep neural network model to classify peptides into unstable (putative degrons) or stable groups (Fig. 2a, Methods). The model takes as features the sequence of a peptide and six of its biophysical properties, which correlated with PSI in the tFT-$X_{12}$ library (Supplementary Fig. 1e), so as to identify sequence-specific degrons for which overall hydrophobicity is not an important determinant. After 30 training epochs, the model achieved an overall training accuracy of 0.85 (Supplementary Fig. 1f, Methods), correctly classified 4110 of the 4726 peptides with PSI < 0.5 in the tFT-$X_{12}$ library as unstable and 37330 of the 41426 peptides with PSI > 0.5 as stable. Next, we used SHAP to interpret the model predictions. The goal of SHAP is to explain the classification of a given peptide as stable or unstable by computing the contribution of each feature (amino acid at each position, biophysical properties per position and of the peptide as a whole) to the prediction, in the form of SHAP values or contribution scores, and in this way understand features of degrons (Fig. 2a, Methods). SHAP contribution scores for the 4110 correctly classified putative degrons showed that the peptide sequence contributed the most to the prediction (Supplementary Fig. 1g). Both positional and global biophysical properties, the aliphatic index in particular, also contributed to the classification.

To identify potential degron motifs, we clustered the 4110 putative degrons based on the vectors of SHAP contribution scores and calculated mean contribution scores for each cluster (Methods). Although SHAP contribution scores can be interpreted for each individual peptide (Fig. 2a), we reasoned that clustering would identify similar peptides and define common features learned by the model that result in their classification as putative degrons, leading to putative motifs. This approach yielded 56 clusters, including 21 clusters

with mean sequence contribution scores suggestive of C-degron motifs (at least one mean sequence contribution score above 0.05 at positions −5 to −1, Methods), containing a total of 1476 peptides (Supplementary Fig. 2a, b). We focused on three clusters containing some of the least hydrophobic degrons in the tFT-$X_{12}$ library (Fig. 2b, Supplementary Fig. 2b), for which the model identified a strong contribution towards instability of a C-terminal asparagine (Fig. 2c). Large hydrophobic amino acids [ILMV] but not [FW] at position −2 also appeared to contribute towards instability in these clusters, with largest contribution scores for [ILV] in clusters 31 and 45 (Fig. 2c). This suggests ΦN as a potential C-degron motif, where Φ = [ILMV]. Further supporting this analysis, the IN, LN, MN and VN dipeptides were enriched at C-termini of the 4726 putative degrons in the tFT-$X_{12}$ library (Fig. 2d). Moreover, IN, LN and VN, but not FN or WN, were among the most destabilizing dipeptides when located at the C-terminus compared to internal positions (Fig. 2e).

We confirmed these observations with tFT-$X_{12}$ libraries where we fixed the IN, LN, MN and VN dipeptides either at the C-terminus or in the middle of otherwise random $X_{12}$ peptides. In all tested cases, the fraction of cells with putative degrons was significantly higher when the peptides ended with ΦN compared to the fully random tFT-$X_{12}$ library or compared to libraries with ΦN in the middle of the peptides (Fig. 2f, Supplementary Fig. 3a). Furthermore, when we fixed only the asparagine at −1 or Φ at −2, the percentage of cells with putative degrons dropped close to background levels (Fig. 2f, Supplementary Fig. 3b). Together, these observations suggest that ΦN is likely a C-degron motif.

## Factors targeting ΦN C-degrons

We sought to identify the machinery that targets proteins with ΦN C-termini for degradation. First, we selected ten $X_{12}$ peptides ending with IN, LN, MN or VN (ΦN peptides) from the 4726 putative degrons in the tFT-$X_{12}$ library and a stable control peptide ($X_{12}$s) (Fig. 3a, Supplementary Fig. 3c). These peptides satisfied at least one of the following criteria: lowest PSI, least hydrophobic or with most sequencing reads (and PSI < 0.5) within the specific C-terminal dipeptide group in the tFT-$X_{12}$ library. In agreement with their low PSIs, strains expressing individual tFT-ΦN constructs exhibited substantially lower mCherry/sfGFP ratios compared to the stable tFT-$X_{12}$s construct or the tFT alone (Fig. 3b, Supplementary Fig. 3d). The tFT readout can vary between subcellular compartments[38]. However, all constructs, including the stable controls, localized to the cytosol and nucleoplasm (Supplementary Fig. 3e), excluding a contribution of the intracellular environment to the observed differences in mCherry/sfGFP ratios. Using cycloheximide (CHX) chase experiments, we found that the ΦN peptides acted as degrons. All tFT-ΦN constructs were degraded once protein translation was blocked with CHX (Fig. 3c, Supplementary Fig. 3f), with the half-life increasing with PSI (Supplementary Fig. 3g). Next, we introduced these constructs, except the less potent LN2, MN1 and MN2 degrons, into an array of 138 single knockout mutants of most non-essential UPS components (UPS array, Fig. 3d)[39] using synthetic genetic array (SGA) methodology for semi-automated crossing[40,41]. Fluorescence measurements of the resulting ordered colony arrays identified Das1 as the only non-essential factor involved in the turnover of four tFT-ΦN constructs: IN2, LN1, LN3 and VN2 (Fig. 3e, Supplementary Fig. 4a, Supplementary Data 2). Both their abundance (sfGFP intensity) and stability (mCherry/sfGFP ratio) were strongly increased in the das1Δ mutant. An independent DAS1 knockout fully stabilized the IN2, LN1, LN3 and VN2 constructs according to the tFT readout (Supplementary Fig. 4b) and in CHX chases (Supplementary Fig. 4c). In contrast, the stable controls were not affected and the IN1, VN1 and VN3 constructs were only partially stabilized in the das1Δ mutant. Deletion of the DAS1 paralog YDR131C had no effect on the stability of the IN1, VN1 and VN3 constructs, both alone and in combination with deletion of DAS1 (Supplementary Fig. 4d).

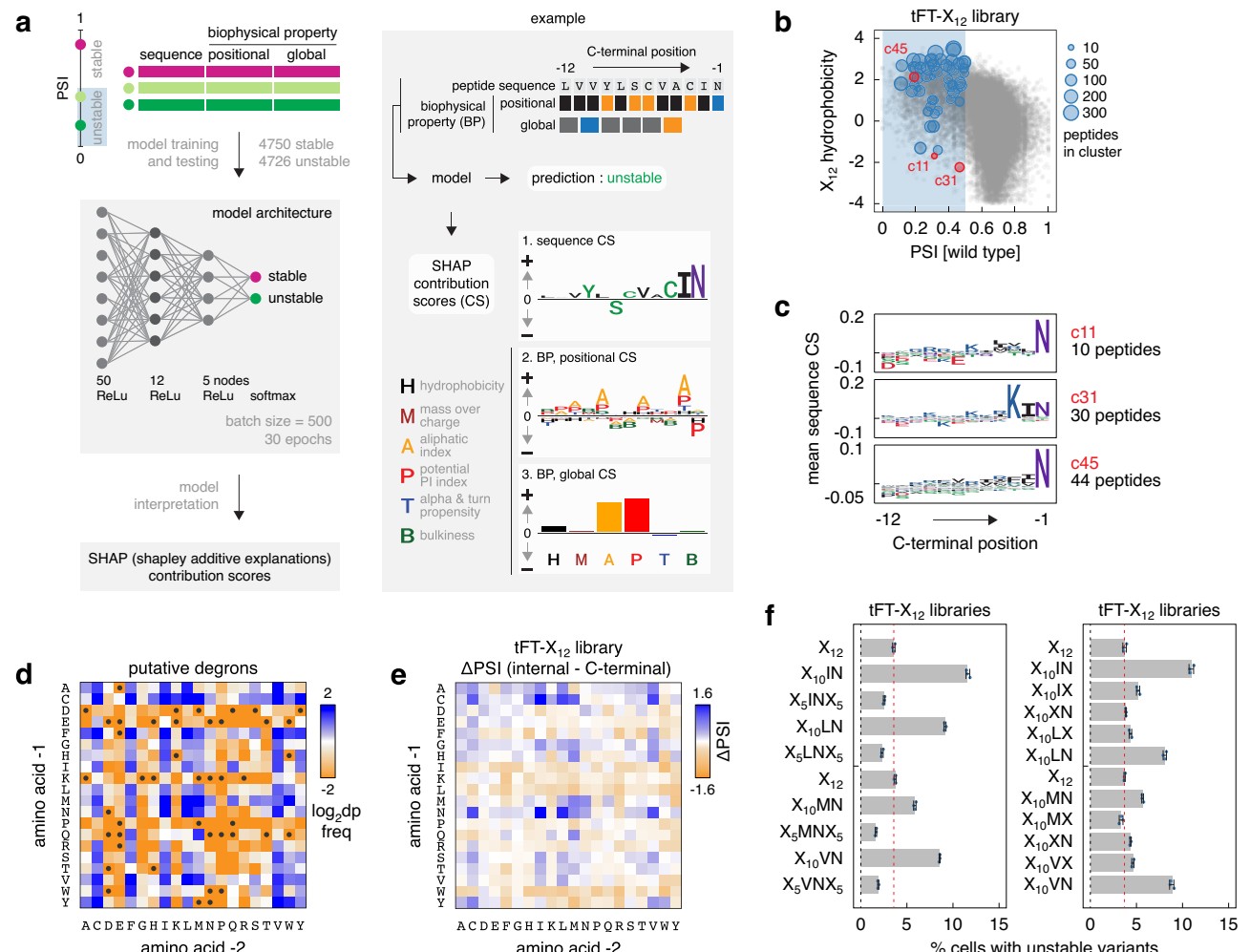

**Fig. 2 | [ILMV]N C-termini promote protein degradation. a** Workflow of the deep neural network model trained on the tFT-X₁₂ library to classify peptides into stable or unstable groups, followed by model interpretation using SHAP contribution scores (CS). The input data, the model output and its interpretation with SHAP are exemplified using one peptide (right panel). SHAP contribution scores for the example peptide are separated by type for clarity (sequence and biophysical properties per position−represented as logos, global biophysical properties represented in a bar chart). **b** Distribution of putative degrons in the tFT-X₁₂ library into 56 groups. K-means clustering of vectors of SHAP contribution scores derived in **a** for the 4110 correctly predicted putative degrons. Each cluster is marked by a representative peptide, with the number of peptides in the cluster indicated by the size of the data point (Methods). All peptides in the tFT-X₁₂ library are shown for comparison (grey). **c** Mean sequence contribution scores for clusters c11, c31 and c45 highlighted in **b**. **d** Relative frequency of dipeptide motifs (dp) at C-termini of 4726 putative degrons from the tFT-X₁₂ library, normalized to the relative frequency in the whole library. Motifs absent from C-termini of the putative degrons are marked with black circles. **e** Heatmap of differences in mean PSI between peptides in the tFT-X₁₂ library with each indicated dipeptide sequence located anywhere between position −12 to −3 (internal) and at positions −2 and −1 (C-terminal). **f** Flow cytometry analysis of pooled yeast libraries expressing tFT-tagged X₁₂ random peptides with the indicated fixed positions. Percentage of cells expressing unstable variants (mCherry/sfGFP <0.5) in each library, mean ± s.d. (*n* = 3, 10⁵ cells per replicate). Source data are provided as a Source Data file.

Das1 is an F-box substrate receptor, conserved in yeasts, of the SCF (Skp1, cullin, F-box) ubiquitin ligase. The SCF works predominantly with the Cdc34 ubiquitin-conjugating enzyme[42–44]. We thus asked whether Cdc34 and the SCF subunits Skp1 and Cdc53 (cullin) are involved in Das1-dependent turnover of tFT-ΦN constructs using the temperature sensitive alleles *skp1-3*, *cdc53-1* and *cdc34-1* of these essential factors[45]. Fluorescence measurements of colonies showed that the IN2, LN1, LN3 and VN2 constructs were strongly stabilized in the *cdc53-1* mutant at the permissive temperature (23 °C) and fully stabilized at the restrictive temperature (37 °C) (Fig. 3f). The *cdc34-1* mutant had no obvious phenotype at 23 °C but strongly stabilized the four tFT-ΦN constructs at 37 °C (Fig. 3g). Using CHX chases, we observed almost complete stabilization of the tFT-IN2 construct in *cdc53-1* and *cdc34-1* mutants at 37 °C, similar to its stabilization in the *das1Δ* mutant (Fig. 3h, Supplementary Fig. 4e). The strong fitness defect of the *skp1-3* strain led to an apparent stabilization even of the

stable tFT-X₁₂s construct at 23 °C, obfuscating the interpretation of tFT-based stability measurements (Supplementary Fig. 4f). Nevertheless, CHX chases showed that the tFT-IN2 construct was clearly stabilized in the *skp1-3* mutant at 37 °C compared to wild type cells (Supplementary Fig. 4g). Taken together, these results indicate that the SCF^Das1 ubiquitin ligase is involved in the turnover of proteins with ΦN degrons.

Next, we established a yeast two-hybrid (Y2H) assay to test interactions between Das1 and ΦN peptides. Exogenous Das1 was tagged with the activation domain (AD) and the 12 amino acid long peptides were fused to the C-terminus of the DNA binding domain (DBD)-DHFR construct. In this assay, expression of the two-hybrid reporters, detected via growth on selective medium, occurs only upon interaction of AD and DBD fusion proteins[46] (Methods). We detected interactions between the IN2, LN1, LN3 and VN2 peptides and Das1 (Fig. 4a). Das1 also interacted with the VN1 peptide, in agreement with

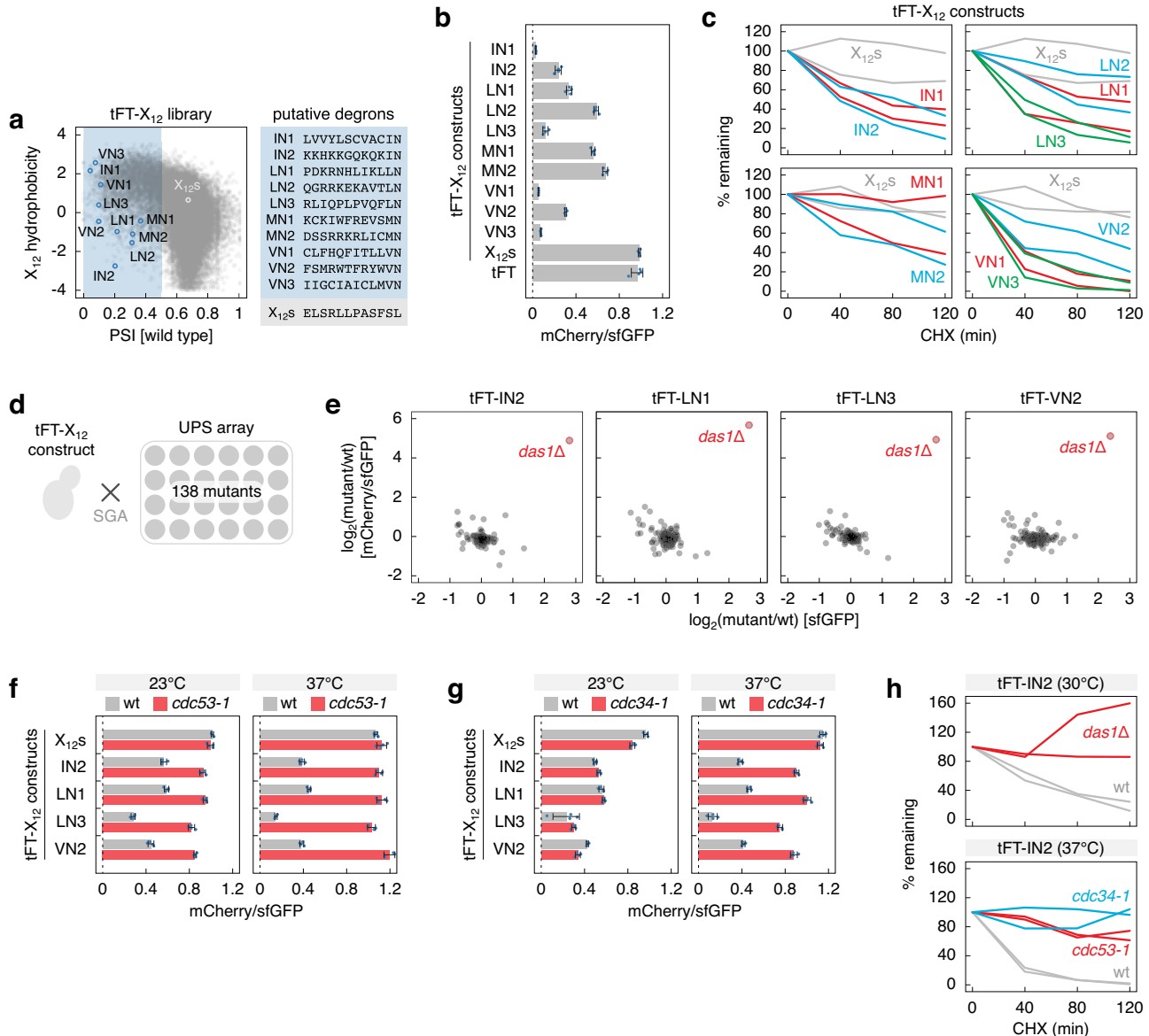

**Fig. 3 | Degradation of proteins with [ILMV]N C-degrons depends on the SCF^Das1 ubiquitin ligase. a** Hydrophobicity, PSI (left) and sequences (right) of selected putative ΦN C-degrons and a control stable peptide (X₁₂s) in the tFT-X₁₂ library. **b** mCherry/sfGFP ratios of colonies expressing tFT-X₁₂ constructs or the tFT alone (mean ± s.d., *n* = 4 independent clones per construct). **c** Degradation of tFT-X₁₂ constructs after blocking translation with CHX. Whole-cell extracts were separated by SDS-PAGE, followed by immunoblotting with antibodies against GFP and Pgk1 as the loading control. For all quantifications of CHX chases hereafter, protein levels of each construct, referenced against Pgk1, were normalized to time point zero. Two replicates per construct. **d** Cartoon of targeted screens to identify UPS factors involved in turnover of tFT-X₁₂ constructs. The UPS array consists of 138 single knockouts of non-essential UPS components (Supplementary Data 2). Each construct is introduced into the UPS array by semi-automated crossing using synthetic genetic array (SGA) methodology, followed by fluorescence measurements of the ordered colony arrays. **e** Screens with tFT-X₁₂ constructs performed according to **d**. Mean fold changes (log₂(mutant/wt), *n* = 4) in sfGFP intensity and mCherry/sfGFP ratio between each mutant and a wild type control (wt, *his3Δ::kanMX*). **f, g** mCherry/sfGFP ratios of colonies expressing tFT-X₁₂ constructs or the tFT alone for comparison (mean ± s.d., *n* = 4 independent clones per construct). Measurements at permissive (23 °C) or restrictive (37 °C) temperatures for the temperature sensitive *cdc53-1* and *cdc34-1* alleles, and the corresponding wild type. **h** Degradation of the tFT-IN2 construct after blocking translation with CHX. The *cdc53-1*, *cdc34-1* and the corresponding wild type strains were shifted to 37 °C for 3 h before adding CHX. Quantification of two replicates per strain and condition. Source data are provided as a Source Data file.

the partial stabilization of the tFT-VN1 construct in the absence of *DAS1* (Supplementary Fig. 4a, b), but not with IN1, VN3 or X₁₂s peptides. This suggests that Das1 is the factor recognizing ΦN degrons.

## Specificity of the SCF^Das1 ubiquitin ligase

We thus sought to understand the specificity of SCF^Das1. First, we asked whether the IN2, LN1, LN3 and VN2 peptides are C-degrons. Using MPS profiling, we tested how adding a single amino acid to the C-terminus (+1 position) of tFT-ΦN constructs affects their turnover. The control

tFT-IN1 construct remained unstable and Das1-independent regardless of the +1 amino acid (Fig. 4b, Supplementary Data 3). Strikingly, almost any +1 amino acid rendered the tFT-IN2 and tFT-LN3 constructs fully stable, suggesting that IN2 and LN3 are C-degrons. In contrast, most amino acids at +1 position did not affect Das1-dependent turnover of tFT-LN1 and tFT-VN2 (Fig. 4b, Supplementary Data 3). Despite these differences, there were trends common to the four Das1 degrons: stabilization by capping the C-terminal ΦN motif with one of six amino acids [DEKRPW]; and persistence of Das1-dependent turnover upon

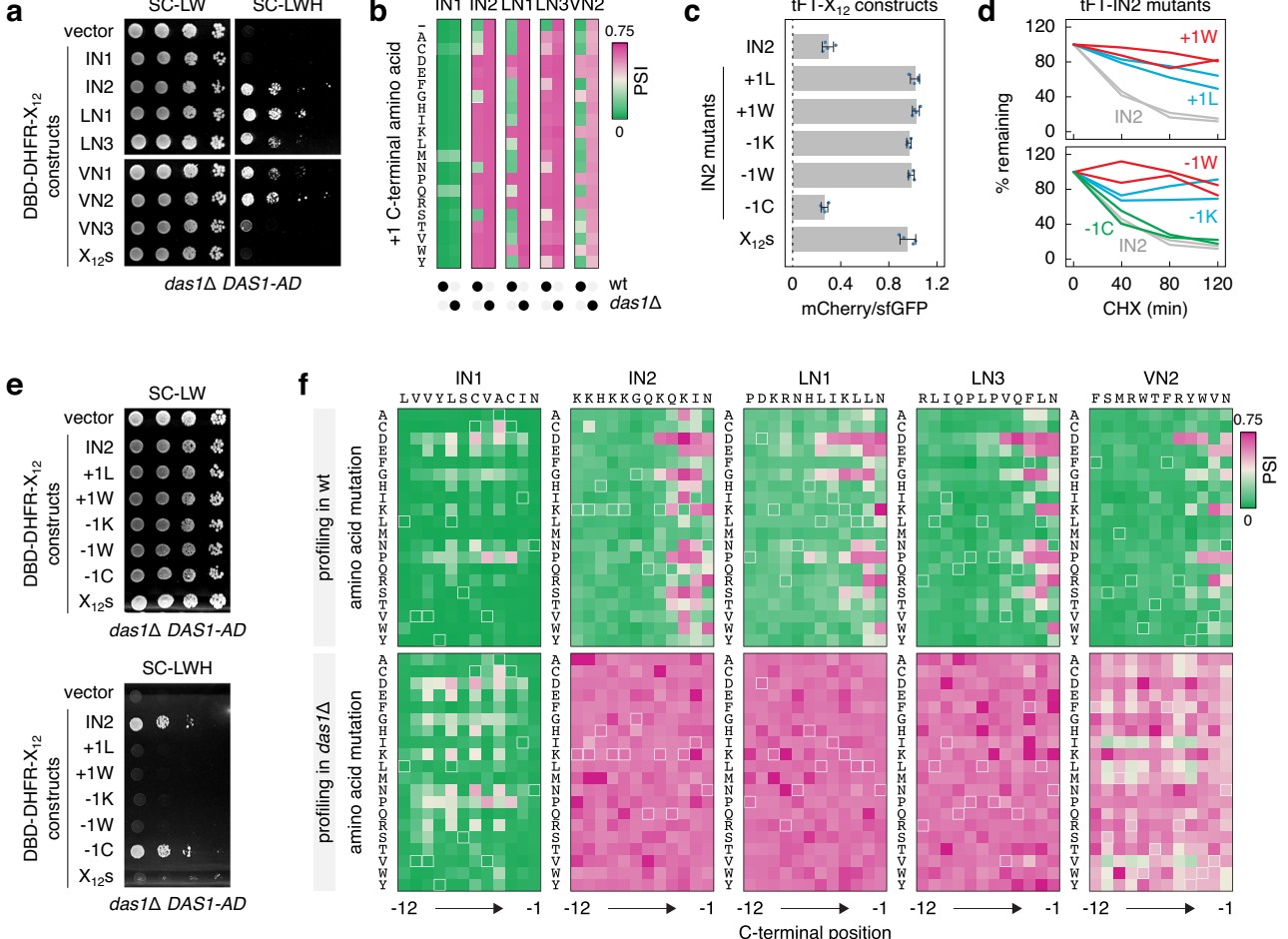

**Fig. 4 | Broad specificity for C-degrons of the SCF$^{Das1}$ ubiquitin ligase. a** Yeast two-hybrid interactions between Das1 fused to the activation domain (AD) and the indicated DNA binding domain (DBD)-DHFR-X$_{12}$ constructs, in the absence of endogenous Das1. Ten-fold serial dilutions on selective media lacking leucine and tryptophan (SC-LW, selection for the AD and DBD plasmids) or, in addition, lacking histidine (SC-LWH). Expression of His3, the two-hybrid reporter (Methods), only occurs upon interaction of AD and DBD fusion proteins. **b** Impact of extending the C-terminus by a single amino acid (+1 position) on the turnover of tFT-X$_{12}$ constructs. Hereafter, heatmaps of PSIs determined by MPS profiling in wild type (wt) and *das1*Δ backgrounds; (-), no C-terminal extension. **c** mCherry/sfGFP ratios of

colonies expressing tFT-X$_{12}$ constructs (mean ± s.d., $n = 4$ independent clones per construct). **d** Degradation of tFT-X$_{12}$ constructs after blocking translation with CHX, quantification of two replicates per construct. **e** Yeast two-hybrid interactions between Das1-AD and DBD-DHFR-X$_{12}$ constructs, as in **a**. **f** Impact of single amino acid mutations on the turnover of tFT-X$_{12}$ constructs. Heatmaps of PSIs determined by MPS profiling of saturation mutagenesis libraries in wild type and *das1*Δ backgrounds. Hereafter, peptide sequences subjected to saturation mutagenesis are indicated above each heatmap and highlighted with light outlines on the heatmaps. Source data are provided as a Source Data file.

addition of one of the four amino acids [ACNS] (Fig. 4b). We verified these observations with three orthogonal assays using the IN2 peptide and two mutants with a +1 leucine (+1 L) or a +1 tryptophan (+1 W). tFT-IN2 was stabilized by +1 L and +1 W mutations both in the tFT assay (Fig. 4c) and in CHX chases (Fig. 4d). Moreover, both mutations abolished the Y2H interaction with Das1 (Fig. 4e). Together, these observations suggest that the ΦN peptides are C-degrons that potentially fall into a broad motif.

Next, we performed saturation mutagenesis of the IN2, LN1, LN3 and VN2 peptides, whereby each amino acid was substituted with all possible amino acids at that position, to understand the sequence requirement for recognition by Das1. MPS profiling of the saturation mutagenesis libraries in wild type and *das1*Δ backgrounds showed that the control tFT-IN1 construct remained generally unstable and Das1-independent irrespective of the mutated position (Fig. 4f, Supplementary Data 4). For the tFT-IN2, -LN1, -LN3 and -VN2 constructs, mutations of amino acids −12 to −6 had largely no effect on their turnover, consistent with the notion that the IN2, LN1, LN3 and VN2 peptides are C-degrons. Multiple amino acids, including charged ones [DEKR] and [GPS], were disallowed at positions −5 to −2. The large

hydrophobic amino acids [ILMV] were preferred compared to [FW] at position −2, consistent with the ΦN motif suggested by the SHAP analysis (Fig. 2c). Notably, mutations of the −1 asparagine to one of the six amino acids [DEKRPW] consistently stabilized the constructs (Fig. 4f, Supplementary Data 4), resembling the trends in the C-terminal capping experiment (Fig. 4b). We verified these observations using the IN2 peptide and three mutants with the −1 asparagine replaced by lysine (−1K), tryptophan (−1W) or cysteine (−1C). tFT-IN2 was stabilized by −1K and −1W mutations both in the tFT assay (Fig. 4c) and in CHX chases (Fig. 4d). Both mutations abolished the Y2H interaction with Das1 (Fig. 4e). In contrast, the −1C mutant exhibited fast turnover, similar to the wild type tFT-IN2 construct (Fig. 4c, d), and interacted with Das1 in the Y2H assay (Fig. 4e). Thus, Das1 appears to have a broad specificity towards C-degrons.

The apparent promiscuity of Das1 is thus far suggested only by the mutagenesis of four ΦN C-degrons. Therefore, we performed a systematic search for Das1 degrons by MPS profiling of a tFT-X$_{12}$ degron library in *das1*Δ, *cdc34-1*, *cdc53-1* and *doa10*Δ mutants (Fig. 5a). This library contained all putative degrons from the tFT-X$_{12}$ library (Fig. 1c) and a set of stable peptides as controls (Supplementary Data 5). The

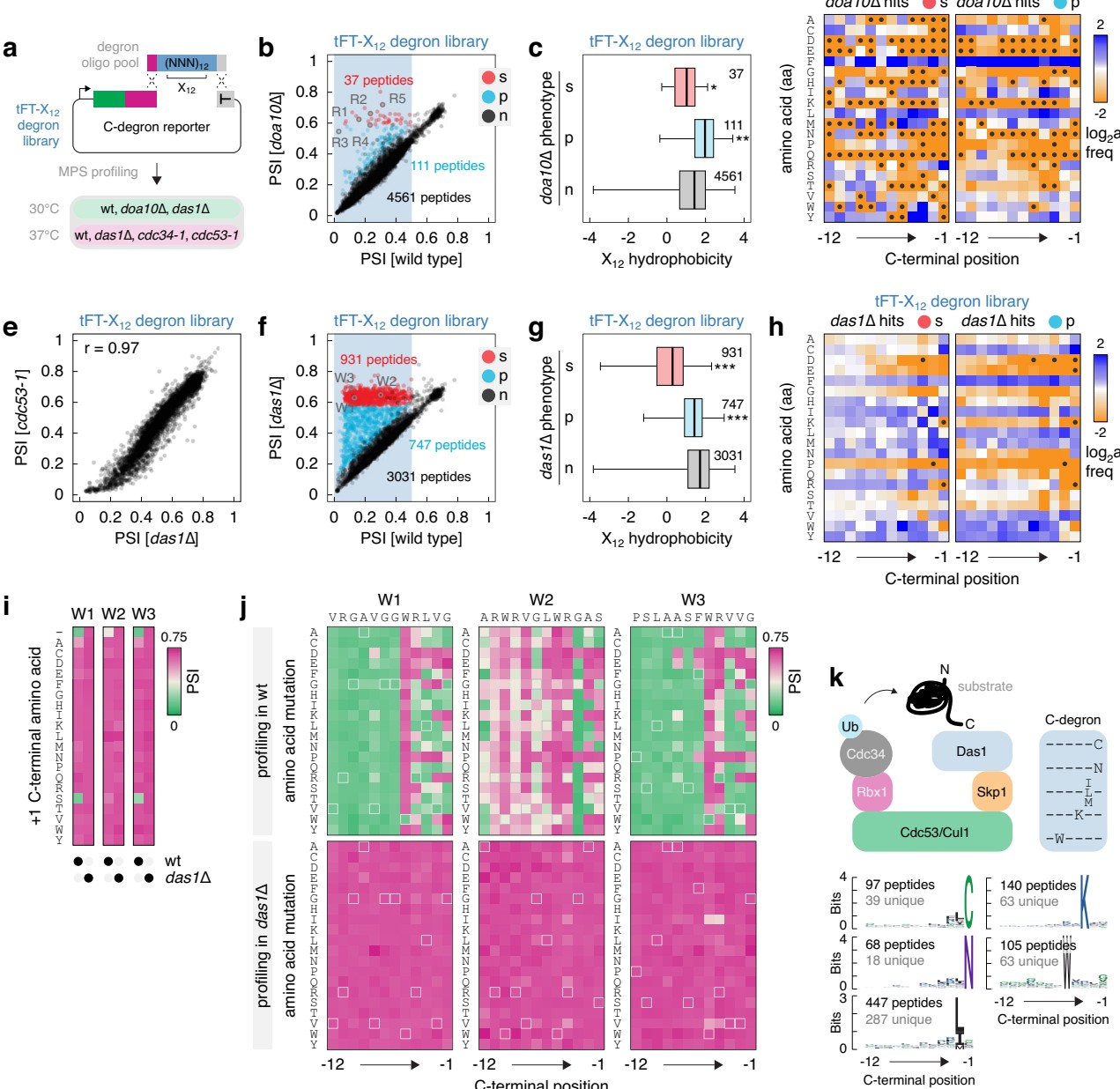

**Fig. 5 | Survey of C-degrons targeted by the SCF$^{Das1}$ ubiquitin ligase. a** Workflow to identify C-degrons targeted by SCF$^{Das1}$. The tFT-X$_{12}$ degron library consists of 4709 putative degrons from the tFT-X$_{12}$ library and 274 stable peptides as controls (Supplementary Data 5). **b**–**d** MPS profiling of the tFT-X$_{12}$ degron library in wild type and *doa10Δ* backgrounds. Constructs stabilized (s), partially stabilized (p) or not affected (n) in the *doa10Δ* mutant. **b** Scatter plot of PSIs, excluding stable controls. R1 to R5, putative Doa10 degrons with a terminal arginine. **c**, Hydrophobicity of the Doa10-dependent constructs. Number of peptides in each group is indicated. Centerlines mark the medians, box limits indicate the 25th and 75th percentiles, and whiskers extend to the minimum and maximum value in each group. \**p* = 0.012, \*\**p* = $4 \times 10^{-10}$ in a one-sided Mann-Whitney *U*-test. **d** Relative amino acid frequency in Doa10-dependent constructs, normalized to the relative frequency in the tFT-X$_{12}$ library (Fig. 1c). Amino acids absent per position in Doa10-dependent constructs are marked with black circles. **e** PSIs in the tFT-X$_{12}$ degron library, excluding stable controls, determined by MPS profiling in *das1Δ* and *cdc53-1* backgrounds. **f**–**h** MPS profiling of the tFT-X$_{12}$ degron library in wild type and *das1Δ* backgrounds.

Constructs stabilized (s), partially stabilized (p) or not affected (n) in the *das1Δ* mutant. **f** Scatter plot of PSIs, excluding stable controls. W1 to W3, putative Das1 degrons with a tryptophan at position −5. **g** Hydrophobicity of the Das1-dependent constructs. Number of peptides in each group is indicated. Centerlines mark the medians, box limits indicate the 25th and 75th percentiles, and whiskers extend to the minimum and maximum value in each group. \*\*\**p* < $2.2 \times 10^{-16}$ in a one-sided Mann-Whitney *U*-test. **h** Relative amino acid frequency in Das1-dependent constructs, normalized to the relative frequency in the tFT-X$_{12}$ library (Fig. 1c). Amino acids absent per position in Das1-dependent constructs are marked with black circles. **i**, **j** Heatmaps of PSIs determined by MPS profiling. Impact of extending the C-terminus by a single amino acid ( +1 position) (**i**) and saturation mutagenesis (**j**) of tFT-W1, tFT-W2 and tFT-W3 constructs from **f**. **k** Model of substrate recognition by SCF$^{Das1}$ via C-degrons (left) and potential C-degron motifs (right). Bottom, sequence logos for the five motifs, based on the 931 Das1 degrons from **f**. Number of peptides forming and unique to each logo are indicated. Source data are provided as a Source Data file.

mutant of the Doa10 ubiquitin ligase, involved in endoplasmic reticulum-associated protein degradation (ERAD)[47,48], was included for comparison. Doa10 targets a variety of hydrophobic degrons, including N-termini of secretory proteins, C-termini of tail-anchored proteins and other hydrophobic sequences with transmembrane domain-like properties[7,26,30,32,49–52].

MPS profiling of multiple replicates per mutant showed good reproducibility (Supplementary Fig. 5a) and the stable control peptides were not affected in any mutant, as expected (Supplementary Fig. 5b). Knockout of *DOA10* completely stabilized only 37 tFT-X$_{12}$ constructs (Fig. 5b, Supplementary Data 5, Methods). These degrons had high overall hydrophobicity, similar to the rest of the unaffected tFT-X$_{12}$ degron library (Fig. 5c) and, consistently, were enriched in the hydrophobic amino acids phenylalanine and leucine (Fig. 5d). Interestingly, Doa10 degrons were enriched in arginine at position −1 (21 out of 37 degrons had a C-terminal arginine, Fig. 5d, Supplementary Fig. 5c). We thus tested the importance of the C-terminal amino acid with five Doa10 degrons (R1 to R4 with R at −1, R5 with R at −2, Fig. 5b). MPS profiling of high-throughput mutagenesis libraries showed that adding a single amino acid to the C-terminus had negligible impact on Doa10-dependent turnover of tFT-tagged R1 to R4 peptides, indicating that these are not C-degrons (Supplementary Fig. 5d, Supplementary Data 3). Nevertheless, the key determinants in each degron were located at positions −5 to −1, whereby hydrophobic residues [FILMV] were preferred at positions −4 to −2 (Supplementary Fig. 5e, Supplementary Data 4). Only constructs with arginine or lysine at −1 exhibited fast Doa10-dependent turnover, suggesting a minimal Doa10 degron motif [FILMV]$_3$[KR]. Interestingly, replacing the −1 arginine with one of three amino acids [ACN] in R1-R4 peptides yielded degrons that were largely Doa10-independent and that conformed to the broad specificity of Das1 (Supplementary Fig. 5e). In contrast, the R5 peptide appeared to be a degron with strict requirements across the C-terminal positions (Supplementary Fig. 5e) and remained a degron but Doa10-independent with most amino acids at the +1 position (Supplementary Fig. 5d).

MPS profiling of the tFT-X$_{12}$ degron library in *das1Δ*, *cdc34-1* and *cdc53-1* mutants showed that the phenotypes of the three mutants were highly correlated (Fig. 5e, Supplementary Fig. 5a, f, Supplementary Data 5). This suggests that the majority of degrons in the library targeted by the SCF ubiquitin ligase are recognized by Das1. This is in stark contrast with observations in human cells, where cullin-RING ubiquitin ligases target a variety of C-degrons using receptor subunits dedicated to different C-degron motifs[22,23]. Remarkably, out of 4709 putative degrons in the library, 1678 (-36%) were Das1-dependent, including 931 (-20%) constructs that were completely stabilized (hereafter referred to as Das1 degrons) and 747 (-16%) constructs that were partially stabilized in the *das1Δ* mutant (Fig. 5f, Supplementary Fig. 5g). The set of Das1-dependent degrons included the IN2, LN1, LN3 and VN2 peptides, which were used in the genetic screens that led to the identification of Das1, and the less potent LN2, MN1 and MN2 degrons (Supplementary Fig. 5h). The phenotype of the *das1Δ* mutant was clearly distinct from the *doa10Δ* mutant, as only two constructs partially stabilized in the *das1Δ* background (FLSRHFLFLSFV and GQRIKWGPIYLT degrons) were also affected in the *doa10Δ* mutant. Partially Das1-dependent degrons tended to be more hydrophobic (Fig. 5g) and, consistently, were enriched in hydrophobic amino acids throughout positions −12 to −1 (Fig. 5h). In contrast, the 931 Das1 degrons were significantly less hydrophobic than the rest of the tFT-X$_{12}$ degron library (Fig. 5g). Moreover, Das1 degrons were depleted of [DEKRPT] and enriched in most other amino acids at position −1, in particular [CMN] (Fig. 5h). This is consistent with the mutagenesis of ΦN C-degrons (Fig. 4), further highlighting the broad specificity of Das1.

We observed similarly strong amino acid enrichments at other positions in Das1 degrons, including [ILM] at position −2, K at −3 and W

at −5 (Fig. 5h). Whereas the preferences at positions −2 and −3 resembled those seen in ΦN C-degrons (Figs. 2c, e and 4), the enrichment of W at −5 raised the possibility of a separate degron motif. We thus analyzed the importance of W at −5 by high-throughput mutagenesis of three degrons, W1, W2 and W3 (Fig. 5f). These peptides had a glycine (W1 and W3) or a serine (W2) at position −1. MPS profiling showed that W1, W2 and W3 are C-degrons, as adding almost any single amino acid to the C-terminus stabilized the corresponding tFT fusions (Fig. 5i). In contrast to ΦN C-degrons (Fig. 4b), addition of only one specific amino acid, S at +1, preserved Das1-dependent turnover of the tFT-W1 and tFT-W3 constructs. Saturation mutagenesis revealed patterns at positions −3 to −1 similar to ΦN C-degrons (Fig. 4f), including the disallowed amino acids [DEKRPW] at position −1 (Fig. 5j). However, W1-W3 degrons had stricter requirements at position −4 and, in contrast to ΦN C-degrons, W at −5 was absolutely required for Das1-dependent turnover, suggesting that W at −5 could be a separate degron motif. Moreover, whereas most Das1 degrons (653 out of 931) can be described by the strongest amino acid enrichments at positions −5 to −1, i.e., 653 degrons had W at −5, K at −3, [ILM] at −2 and/or [CN] at −1 (Fig. 5k), only 21 degrons had 3 or more of these preferred amino acids simultaneously (Supplementary Fig. 5i). Together, these observations support the idea of multiple distinct but potentially overlapping degron motifs recognized by Das1.

## Substrates and functions of SCF^Das1

Having established the broad specificity of Das1, we sought to identify endogenous substrates of SCF^Das1. Deletion of *DAS1* does not appear to affect the abundance or turnover of the yeast proteome, at least in a haploid reference strain grown on glucose as carbon source[17,53], suggesting that Das1 substrates could be conditional. Therefore, we first surveyed the C-termini of all yeast proteins (C-terminome) for Das1 degrons (Fig. 6a). MPS profiling of a yeast C-terminome library, consisting of 12 amino acid long C-termini of 6793 annotated yeast open reading frames (ORFs) fused to the tFT, identified 458 putative degrons (Fig. 6b, Supplementary Fig. 6a, Supplementary Data 6). These sequences were on average more hydrophobic than the rest of the C-terminome (Supplementary Fig. 6b). Remarkably, 246 (-54%) of these degrons appeared to be targeted by Das1, including 198 (-43%) that were completely Das1-dependent (Das1 degrons) (Fig. 6b). These 198 Das1 degrons had properties similar to the Das1 degrons in the tFT-X$_{12}$ degron library: lower overall hydrophobicity compared to the other yeast C-degrons (Fig. 6c), depletion of proline, tryptophan and charged amino acids at position −1 and enrichment of W at −5, K at −3, [IL] at −2 and [CN] at −1 (Fig. 6d).

Approximately 25% of Das1 degrons in the yeast C-terminome originated from dubious ORFs, which are unlikely to encode functional proteins (Supplementary Fig. 6c), and the remaining degrons corresponded to proteins with a variety of different subcellular localizations and functions (Supplementary Fig. 6d). Considering that Das1 localizes to the cytosol[54], we hypothesized that it might be involved in protein quality control and in this way target for degradation proteins that are otherwise destined to different compartments.

To assess this hypothesis, we analyzed the role of Das1 in turnover of full-length yeast proteins that contain Das1 C-degrons. We expressed each ORF N-terminally tagged with an mNeonGreen-mCherry timer[55,56] at endogenous levels or overexpressed from a strong constitutive promoter, which could lead to production of unnecessary (e.g., orphan subunits of protein complexes) or abnormal molecules (e.g., misfolded or mislocalized)[3,4,57,58]. When expressed at endogenous levels, none of the tested potential Das1 substrates exhibited Das1-dependent turnover (Fig. 6e). In contrast, 18 out of 74 overexpressed tFT fusions were stabilized in the absence of *DAS1*, as indicated by increased mCherry/mNeonGreen ratios in the *das1Δ* mutant (Fig. 6f). This is consistent with the idea of Das1 as a protein quality control factor. Notably, 12 of these Das1 substrates were subunits of protein

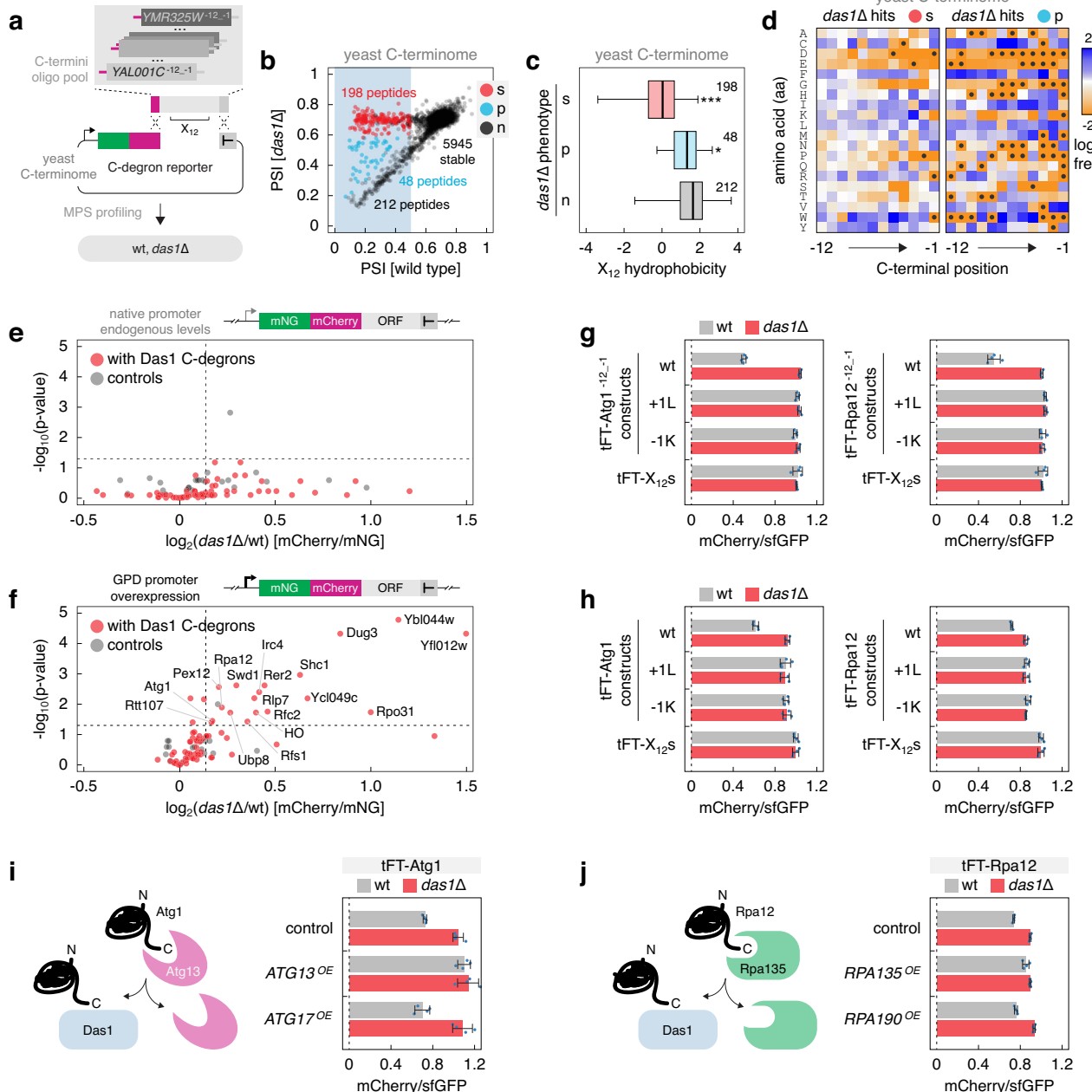

**Fig. 6 | Substrates of the SCF^Das1 ubiquitin ligase. a** Workflow to identify yeast C-degrons targeted by SCF^Das1. The yeast C-terminome library consists of 12 amino acid long C-termini from 6793 yeast ORFs (Supplementary Data 6). **b** PSIs in the C-terminome library determined by MPS profiling in wild type and *das1Δ* backgrounds. Within unstable constructs (blue region, PSI [wild type] <0.5), those stabilized (s), partially stabilized (p) or not affected (n) in the *das1Δ* mutant are indicated. **c** Hydrophobicity of the C-terminome constructs stabilized in the *das1Δ* mutant from **b**. Number of peptides in each group is indicated. *p = 0.012, ***p < 2.2 × 10⁻¹⁶ in a one-sided Mann-Whitney *U*-test. Centerlines mark the medians, box limits indicate the 25th and 75th percentiles, and whiskers extend to the minimum and maximum value in each group. **d** Relative amino acid frequency in Das1-dependent constructs from **b**, normalized to the relative frequency in the C-terminome library. Left, stabilized (s) and right, partially stabilized (p) constructs in the *das1Δ* mutant. Amino acids absent per position in Das1-dependent constructs are marked with black circles. **e, f** Turnover of full-length yeast proteins corresponding to the Das1 degrons (stabilized group in **b**). mNeonGreen (mNG)-

mCherry-tagged proteins were expressed from native promoters (**e**) or over-expressed from the *GPD* promoter (**f**). Proteins lacking Das1-dependent degrons at the C-terminus were analyzed as controls. Mean fold changes in mCherry/mNG ratios of colonies (log₂(*das1Δ*/wt), n = 4). Proteins significantly stabilized in the *das1Δ* mutant (*p*-value < 0.05 in a two-sided *t*-test (adjusted for multiple testing) and *das1Δ*/wt [mCherry/mNG] > 1.1) are indicated (**f**). **g, h** mCherry/sfGFP ratios of colonies expressing tFT-tagged C-termini of Atg1 or Rpa12 (**g**) or tFT-tagged full-length Atg1 or Rpa12 (**h**). Wild type C-termini or proteins (wt), with an additional C-terminal leucine (+1 L) or with the C-terminal amino acid mutated to lysine (−1K). The stable tFT-X₁₂s construct was analyzed for comparison. Mean ± s.d., n = 4 independent clones per construct. **i, j** mCherry/sfGFP ratios of colonies expressing tFT-tagged full-length proteins without (control) or with overexpression (OE) of the indicated factors. Measurements normalized to the stable tFT-X₁₂s construct in the corresponding control or OE backgrounds. Mean ± s.d., n = 4 independent clones per construct. Left, models of Das1-dependent degradation of orphan Atg1 (**i**) and Rpa12 (**j**). Source data are provided as a Source Data file.

complexes (Supplementary Fig. 6e), suggesting that SCF$^{Das1}$ is involved in degradation of orphan protein complex subunits.

We focused on two proteins to test this possibility, the autophagy kinase Atg1 and the RNA polymerase I subunit Rpa12. The C-terminal domains of both proteins are involved in protein-protein interactions within complexes: the C-terminal domain of Atg1 binds to Atg13[59], whereas Rpa12 makes contacts with Rpa190, Rpa135, Rpa49 and Rpa34 and its C-terminal domain can interact with the polymerase I active site[60]. First, we analyzed the 12 amino acid long C-termini of Atg1 (Atg$^{-12,-1}$) and Rpa12 (Rpa12$^{-12,-1}$) in isolation using high-throughput mutagenesis to identify degron-disrupting mutations. MPS profiling showed that Atg$^{-12,-1}$ and Rpa12$^{-12,-1}$ are C-degrons, as every amino acid added to the C-terminus stabilized the corresponding tFT fusions (Supplementary Fig. 6f). Saturation mutagenesis revealed patterns consistent with the broad specificity of Das1, including faster turnover of tFT-Rpa12$^{-12,-1}$ with F at position −5 mutated to W (Supplementary Fig. 6g). We verified these observations with individual constructs using two mutants with a +1 leucine ( +1 L) or the −1 asparagine replaced by lysine (−1K) (Fig. 6g). Notably, introducing these mutations into full-length Atg1 and Rpa12 stabilized both tFT-tagged proteins to the same extent as deletion of *DAS1* (Fig. 6h). This indicates that Atg1 and Rpa12 are recognized by Das1 via their C-termini.

Atg1 and Rpa12 undergo Das1-dependent turnover only when overexpressed (Fig. 6e, f) and likely in excess relative to their binding partners. This suggests that Das1 recognizes orphan Atg1 and Rpa12 molecules, which failed to assemble into complexes. If this model is correct, Das1-dependent turnover of Atg1 and Rpa12 should be abolished upon overexpression of their binding partners. Indeed, overexpression of Atg13 but not Atg17 stabilized tFT-Atg1 and no further stabilization was detected upon deletion of *DAS1* (Fig. 6i). Similarly, Das1-dependent turnover of tFT-Rpa12 was abolished by overexpression of Rpa135 (Fig. 6j). Therefore, we conclude that the SCF$^{Das1}$ ubiquitin ligase is a broad-specificity quality control factor that recognizes orphan complex subunits via C-degrons.

## Discussion

Despite the general importance of selective protein degradation, the specificity of most ubiquitin ligases remains poorly understood. Approaches to systematically identify degrons and the corresponding ubiquitin ligases, such as those applied herein, hold the promise of filling this knowledge gap to reveal the organization and specificity of the ubiquitin-proteasome system.

Our survey of C-terminal degrons using libraries of 12 amino acid long peptides revealed that ~10% of random peptides (4726 out of 46152 sequences in the tFT-X$_{12}$ library) and ~7% of yeast C-termini (458 out of 6403 sequences in the yeast C-terminome library) encode degrons. It is possible that additional degrons could be identified using reporters including unstructured regions of different lengths and sequence composition as initiation sites of proteasomal degradation[28,29,61]. The occurrence of ubiquitin-independent degrons in these libraries, as recently uncovered in human cells[62], remains to be determined. Nonetheless, the degrons identified herein are enriched in hydrophobic amino acids and many are likely recognized not in a sequence-specific manner but based on their overall high hydrophobicity, as seen in quality control of misfolded proteins or with degrons recognized by the Doa10 ubiquitin ligase[2,7,30,36]. And yet, our analysis showed that not only does a significant fraction of these degrons appear to be sequence-specific, but they are also recognized by a single ubiquitin ligase, SCF, using the F-box substrate receptor Das1. It is striking that ~36% of random degrons (1678 out of 4709) and ~54% of degrons in the yeast C-terminome (246 out of 458) are C-degrons targeted by SCF$^{Das1}$, arguing that the SCF/C-degron pathway has a broad specificity towards C-degrons. Further supporting this notion, Hasenjäger et al. independently identified SCF$^{Das1}$ as a factor targeting a wide range of C-degrons[63].

This is in stark contrast with other C-degron pathways found in human cells, where different degron motifs are recognized by interchangeable substrate receptors of several cullin-RING ubiquitin ligases[22,23,64,65]. Among them, Cul2-based complexes recognize multiple related C-degron motifs ending with glycine or diglycine. Interestingly, both glycine and diglycine motifs are depleted from C-termini of eukaryotic proteomes, which led to the suggestion that recognition of such degrons by ubiquitin ligases has shaped eukaryotic proteomes throughout evolution[23,24].

Consistent with the absence of Cul2 in yeast[25], our analysis did not identify glycine C-degron motifs in *S. cerevisiae*. Das1 does recognize degrons that end with a glycine, including ΦN degrons with the C-terminal asparagine mutated to glycine. However, in contrast to similar analyses in human cells[22,23], random degrons in the tFT-X$_{12}$ library were not enriched in glycine at position −1, suggesting that C-terminal glycine is not a strong degron motif in yeast. Furthermore, human glycine C-degrons are inactive in yeast, at least in the context of reporters[30]. Nevertheless, glycine and diglycine motifs are depleted from C-termini of yeast proteins, similarly to other eukaryotic proteomes[23,24], raising questions about the reasons for this depletion. In this respect, it is worth considering that a key feature of ubiquitin and ubiquitin-like (Ubl) proteins is a C-terminal tail ending with a glycine or a diglycine motif that is used for activation and conjugation of Ubls[66]. We thus hypothesize that depletion of glycine motifs, and the diglycine motif in particular, from C-termini in budding yeast and other eukaryotic proteomes arose at least in part from the pressure to minimize crosstalk with the ubiquitin and other Ubl systems.

Compared to other ubiquitin ligases targeting C-degrons, Das1 appears to have an exceptionally broad specificity, determined by amino acids at positions −5 to −1. This notion is supported by two observations: the broad spectrum of random degrons and yeast C-terminal degrons targeted by Das1 and that most single amino acid substitutions at positions −5 to −1 have no impact on Das1 degrons. For instance, over 75% of single mutations at positions −5 to −1 in the LN3 degron had little to no impact on Das1-dependent turnover of the tFT-LN3 fusion. The broad specificity of Das1 towards C-degrons is reminiscent of the broad specificity of Ubr1 towards N-degrons. However, Ubr1 is a large ubiquitin ligase (225 kDa) that recognizes a variety of N-degrons and internal degrons using three distinct substrate-binding sites[10,67]. It remains to be determined whether one or multiple substrate-binding sites in the substantially smaller Das1 (77 kDa) are responsible for its broad specificity.

Our survey of Das1 degrons and saturation mutation of individual degrons identified amino acids strongly disfavored in Das1 degrons, including charged amino acids, proline and tryptophan [DEKRPW] at position −1, and negatively charged amino acids, glycine and proline [DEGP] at positions −5 to −2. These preferences are consistent with prior work that identified Hac1u as a Das1 substrate[68]. Hac1 is a key transcription factor in the unfolded protein response (UPR)[69]. When UPR is inactive, the *HAC1* mRNA remains unspliced and in a largely translationally inactive form. Nevertheless, occasional translation of the unspliced *HAC1* mRNA results in production of an active Hac1u transcription factor, which can lead to unwarranted activation of the UPR. To prevent this, yeast cells employ SCF$^{Das1}$ to recognize and degrade Hac1u via a 10 amino acid C-terminal tail, AVITMTRKLQ, encoded by the *HAC1* intron. Notably, among the spontaneous mutations impairing degradation of Hac1u, which led to the identification of Das1, were several mutations in the Hac1u C-degron[68]. These included truncations and extensions of the C-terminal tail, and mutations at position −1 to K, −2 to P and −5 to P, in agreement with the Das1 preferences revealed by our analyses.

While it is possible that the specificity of Das1 is best reflected by a negative degron motif, dominated by disfavored amino acids, our analysis also revealed clear preferred amino acids at positions −5 to −1 in Das1 C-degrons, consisting of W at position −5, K at −3, [ILM] at −2

and [CN] at −1. As these preferred amino acids rarely cooccurred in over 1100 Das1 degrons, we hypothesize that Das1 recognizes multiple but potentially overlapping C-degron motifs. Dissecting the biophysical properties and structural features of Das1 degrons should help understand the molecular determinants governing the specificity of Das1.

Besides its role in degradation of products from unwanted and rare translation of the unspliced *HAC1* mRNA[68], we show that SCF^Das1 targets for degradation orphan subunits of protein complexes. Consistent with the notion of Das1 as a quality control factor, the 18 Das1 substrates identified herein are enriched in subunits of complexes that function in a variety of different processes, including autophagy, peroxisome biogenesis, transcription, translation, redox metabolism and regulation of gene expression. Our results argue that C-termini of Atg1 and Rpa12 are hidden from Das1 by their complex interaction partners, thus explaining the specificity of Das1 towards orphan Atg1 and Rpa12 and expanding the spectrum of ubiquitin ligases implicated in orphan quality control. Considering that *DAS1* exhibits synthetic sick interactions with components of the splicing, tRNA biogenesis and translation machineries[70], we speculate that Das1 is a factor with extensive functions in removal of aberrant proteins and products of faulty translation, relying on its broad specificity to recognize a variety of C-terminal sequences. As the SCF ubiquitin ligase is among the most ancient CRLs, with functions in cell cycle control conserved from yeast to mammals[71,72], it is possible that the SCF/C-degron pathway or some of its features, such as its sequence determinants or functions in proteostasis, are conserved across eukaryotes. Supporting this possibility, the human SCF, together with the F-box substrate receptor FBXO31, appears to function as a quality control factor by targeting for degradation proteins carrying C-terminal amide degrons formed during oxidative stress[64]. Further work is needed to explore the specificity and targeting of C-degrons by SCF complexes across model organisms, towards a comprehensive understanding of eukaryotic C-degron pathways.

## Methods

### Yeast strains and growth conditions

All yeast strains used in this work are listed in Supplementary Table 1 and are derivatives of BY4741, Y8205, Y7092, ESM356-1, or PJ69-4A. All plasmids used in this work are listed in Supplementary Table 2. Key primers, oligonucleotide pools and degenerate oligonucleotides are described in Supplementary Data 7. Yeast genome manipulations (gene tagging and gene deletion) were performed using PCR targeting and lithium acetate transformation[73–75]. High-throughput N-terminal tagging was performed with the SWAp-Tag approach using the N-SWAT library[54,76] and the donor plasmids pKEK431, pKEK432, pKEK433 and pKEK434 (Supplementary Table 2). Unless stated otherwise, yeast strains were grown at 30 °C in synthetic complete (SC) medium lacking histidine (SC-His) to select for constructs based on the C-degron reporter plasmid pNS002[77] and with 2% (w/v) glucose as carbon source.

### Multiplexed protein stability profiling

Analysis of tFT-peptide libraries by MPS profiling involved construction of libraries, fluorescence-activated cell sorting based on the tFT readout, DNA extraction from the sorted populations and preparation of amplicon libraries, followed by deep sequencing of the amplicon libraries and analysis of the sequencing data.

The C-degron reporter was constructed with a tFT designed specifically for studying C-degrons, in which the slower maturing mCherry is proximal and the faster maturing sfGFP is distal to the degron, such that incomplete proteasomal degradation of sfGFP does not affect the tFT readout[55,77]. The C-terminus of mCherry (RAEGRHSTGGMDELYK) is separated from the assayed peptides by the short GGGS linker. tFT-peptide libraries were constructed by homologous recombination in

yeast using degenerate oligonucleotides or oligonucleotide pools[26,77]. Degenerate oligonucleotides were designed with 42 nucleotide overhangs, homologous to the insertion site in the vector pNS002, flanking the variable region. For each library, complementary degenerate oligonucleotides containing the required number of degenerate positions were annealed at 20 μM total concentration in a total volume of 40 μL. Oligonucleotide pools were designed with 20 nucleotide overhangs and amplified by PCR using 0.8 pmol of the pool as template, primers Cdeg_MPS_extend_F/Cdeg_MPS_extend_R (Supplementary Data 7) and 10 PCR cycles to extend the overhangs to 42 nucleotides. Annealed oligonucleotides (whole annealing volume) or amplified oligonucleotide pools (whole PCR reaction volume) were then transformed together with 15 μg of the pNS002 vector, linearized with EcoRV, into ~2.4 × 10^9 competent yeast cells with the desired genotype. For both types of libraries, transformed cells were grown for 2 days to saturation (optical density at 600 nm $OD_{600} > 7$), and stored as glycerol stocks at −80 °C.

For sorting, libraries were inoculated from frozen stocks and grown overnight to saturation, diluted into fresh medium to 0.2 $OD_{600}$ and grown to 0.8 $OD_{600}$. Fluorescence-activated cell sorting was performed on a FACSAria III SORP cell sorter (BD Biosciences) with a 70 μm nozzle (70 psi) using a 561 nm laser for mCherry excitation, a 600 nm long pass mirror and a 610/20 nm band pass filter for mCherry detection, a 488 nm laser for sfGFP excitation, a 505 nm long pass mirror and a 530/30 nm band pass filter for sfGFP detection. For each library, 5 × 10^6 sfGFP-positive cells were sorted according to the mCherry/sfGFP ratio into 8 stability bins of approximately equal width on the mCherry/sfGFP scale (Supplementary Fig. 7a–d).

For each bin, the sorted cells were grown for 48 h to saturation and ~2 × 10^8 cells were harvested by centrifugation for DNA extraction. DNA was extracted with the QIAprep spin miniprep kit (27106, Qiagen). First, cells were lysed in 250 μl of buffer P1 by vortexing for 3 minutes in the presence of glass beads (G8772, Sigma-Aldrich). Then, 250 μl of buffer P2 followed by 350 μl of buffer N3 were added to the lysate. Subsequent DNA binding to the spin column, washing and elution were performed according to manufacturer's instructions.

The extracted DNA was used as template to amplify the variable region by a two-step PCR strategy using a high-fidelity DNA polymerase (NEBNext ultra II Q5 master mix, M0544L, NEB). The first PCR of 22–24 cycles was performed using primers #232 to #273 (Supplementary Data 7) to obtain products of 133–149 bp. These phased primers contained between 0 and 10 random nucleotides at their 5′ ends to increase library complexity and improve sequencing quality. In addition, the reverse primer had a unique barcode for each stability bin (fraction barcode). For each library, the 8 PCR products, one for each stability bin, were gel-purified (QIAquick gel extraction kit 28706, Qiagen) and pooled proportionally according to the number of cells FACS-sorted into each fraction. This pool was then used as template for the second PCR of 6 cycles with primers P5_Scriptseq and P7_Scriptseq (Supplementary Data 7) to add the P5 and P7 adaptors, yielding products of 255–271 bp. The reverse primer also carried a sample barcode to identify the specific library or genetic background. After gel purification, the final PCR products from multiple libraries were typically combined and sequenced on a NextSeq 500 system (Illumina) in 2 × 80 bp paired-end mode.

Analysis of sequencing data was performed based on a previously described workflow[26], but allowing for a flexible amplicon design with respect to the length and sequence of the variable region, flanking common sequences, barcodes of stability bins and sample barcodes. The applied analysis pipeline is available at https://github.com/Khmelinskii-Lab/Das1_C-degrons/tree/main/NGSpipe2go-MPSprofiling. After sample demultiplexing and generation of FastQ files using Illumina index reads with bcl2fastq conversion software (v2.20, Illumina), raw data quality of FastQ files was checked using FastQC software (v0.11.9)[78]. Samples were checked for potential DNA

contamination from common sources like human, mycoplasma and *E. coli* using FastQ Screen (v0.13)[79]. Cutadapt (v4.0)[80] was used to trim adapter sequences and for NextSeq-specific 3′ end quality trimming applying a threshold of 20. Reads were filtered for a minimum length of 15 nucleotides and a maximum expected number of errors equal to 1 computed from quality values as described[81] using the cutadapt parameter --max-expected-errors 1. Read pairs were assembled using the paired-end read merger (PEAR, v0.9.11)[82] to reconstruct amplicon sequences based on a minimum of 8 bases overlap of the paired reads. Unassembled read pairs were discarded. The variable region and the fraction barcode were subsequently extracted from the assembled sequences by applying a design-specific regular expression with the extract command in UMI-Tools (v1.1.2)[83] using the flanking common sequences. Extracted sequences were added to the read names in the respective FastQ files. Sequences with invalid barcodes were discarded. Reads were counted for each observed combination of variable region and fraction barcode, and were exported as a count matrix in text format. Reading and downstream processing of count matrices was done using the R programming language (v4.1.2)[84]. Protein stability indices (PSIs) per sequence variant were calculated from count data as described[26]. Briefly, PSIs were determined either on read counts per nucleotide sequence or on read counts per translated amino acid sequence generated with the Biostrings R package. First, the observed read counts per bin for each sequence variant were normalized by the summed read counts for all sequence variants of this bin and by the corresponding cell fraction count. The PSI was then calculated as the sum of these values over all bins multiplied with the corresponding cell fraction indices divided by the sum over all bins.

Downstream analysis and data visualization were performed in R and the analysis pipeline is available at https://github.com/Khmelinskii-Lab/Das1_C-degrons. Only sequence variants with more than 10 reads across all stability bins were considered. For the tFT-$X_{12}$ library, all analysis was performed using replicate 1, in which 46152 unique peptides were identified with more than 10 sequencing reads. For the tFT-$X_{12}$ degron library and the yeast C-terminome library, technical replicates were combined using linear regression in the limma R package[85], with false discovery rate (FDR) for each peptide calculated by the Benjamini-Hochberg procedure.

Doa10- or Das1-dependent peptides in the tFT-$X_{12}$ degron library were defined as follows:

stabilized (s): PSI [wt] <0.5 & PSI [mutant] > 0.58 & PSI [mutant] − PSI [wt] > 0.1; partially stabilized (p): PSI [wt] < 0.5 & PSI [mutant] < 0.58 & PSI [mutant] − PSI [wt] > 0.1; not affected (n): PSI [mutant] − PSI [wt] > 0.1 & p-adj > 0.05 | PSI [mutant] − PSI [wt] < 0.1

Das1-dependent peptides in the yeast C-terminome library were defined similarly:

stabilized (s): PSI [wt] < 0.5 & PSI [mutant] > 0.58 & PSI [mutant] − PSI [wt] > 0.1; partially stabilized (p): PSI [wt] < 0.5 & PSI [mutant] < 0.58 & PSI [mutant] − PSI [wt] > 0.1; not affected (n): PSI [wt] < 0.5 & PSI [mutant] − PSI [wt] > 0.1 & p-adj > 0.05 | PSI [wt] < 0.5 & PSI [mutant] − PSI [wt] < 0.1

The PSI threshold separating stable from unstable peptides in mutant backgrounds was higher than in wild type libraries to increase the stringency of the analysis. Yeast C-terminome sequences were mapped to the corresponding open reading frame information taken from Uniprot[86] on 03/07/2023 and filtered for Swiss-Prot reviewed *S. cerevisiae* proteins. Classification of 6611 yeast open reading frames by type (verified, uncharacterized and dubious) and the corresponding slim gene ontology terms were retrieved from YeastMine[87] on 03/07/2023 and 07/07/2023, respectively.

### Deep neural network model and interpretation

An interpretable deep learning approach was used to identify features of degrons in the tFT-$X_{12}$ library.

First, a PSI threshold of 0.5 was set to split the library into unstable (PSI < 0.5) and stable (PSI > 0.5) peptides. This threshold was chosen based on the distribution of PSIs in the library, which is unimodal, centered at 0.668 ± 0.046 (median ± median absolute deviation (mad)) and skewed towards low PSIs. Most constructs in the library are likely stable as the distribution of mCherry/sfGFP ratios in the library is close to a strain expressing only the tFT (Fig. 1b). A dispersion d = 3.7 x mad was chosen based on this distribution of PSIs. With this measure of dispersion, ~4-5% of the PSIs are expected to be outside of the median ± d range for a symmetric distribution. For the experimental distribution, 2% of PSIs are above the median + d threshold and 10% of PSI are below the median - d threshold (Supplementary Fig. 1d).

Then, 4726 unstable and 4750 randomly selected stable peptides (PSI > 0.5) were chosen from the MPS profiling of the tFT-$X_{12}$ library, replicate 1. These peptides were represented by 318 features comprising sequence information, biophysical properties per position and global biophysical properties. Biophysical properties were calculated with the Peptides R package[88]. Based on their high correlations with PSI in the tFT-$X_{12}$ library (Supplementary Fig. 1e), six biophysical properties were used in the model: hydrophobicity[89], aliphatic index[90], bulkiness, alpha and turn propensity[90,91], potential protein interaction index[92] and mass over charge.

The dataset was divided into (a) a training set, consisting of 3750 each of unstable and stable peptides, (b) a validation set, consisting of 20% of the training set taken at random and (c) 4 test sets with peptides excluded from the training set. Test sets 1–3 had no peptides in common. Test set 4 was formed by combining test sets 1–3. These sets were used to check the model performance in seen data (training and validation) as well as unseen data (test sets). The four test sets were used to ensure that assessment of the model performance is not affected by the selection of the test set.

The parameters for the fully connected neural network were chosen by the brute-force method. The neural network had 5 layers, with 3 internal layers of 50, 12 and 5 nodes. Each internal layer had a dropout of 0.3 and a rectified linear unit (reLU) activation function. The second and fourth internal layers had L1 and L2 regularizers of 0.001. The output layer had a softmax activation function. The model had binary cross-entropy as a loss function, Adam optimizer, batch size of 500 and runs in 30 epochs. The performance of the model was measured by five performance metrics after 30 epochs: loss, accuracy, area under curve (AUC), recall, precision and mean squared error (MSE). The model performed consistently across all the sets, with a training accuracy and AUC of 0.85 and 0.93, respectively (Supplementary Fig. 1f).

The model performance was also assessed on replicate 2 of the tFT-$X_{12}$ library (Supplementary Fig. 1a). 47876 peptides with at least 10 sequencing reads were detected in replicate 2, 4837 unstable (PSI < 0.5) and 43039 stable. The model correctly classified 4212 unstable and 38396 stable peptides. The performance metrics were loss = 0.30, accuracy = 0.89, AUC = 0.94, precision = 0.89, recall = 0.89 and MSE = 0.09.

In addition, replicates 1 and 2 were combined using linear regression in the limma R package[85]. Out of 43095 peptides in the combined dataset, 3973 of the 4412 unstable (mean PSI < 0.5) and 34591 of the 38683 stable peptides were correctly classified by the model. The performance metrics were loss = 0.29, accuracy = 0.89, AUC = 0.95, precision = 0.89, recall = 0.89 and MSE = 0.09.

To infer the features important for prediction by the deep neural network, a combinatorial inferential approach, SHapley Additive exPlanations or SHAP[37], was used on the 4110 correctly predicted unstable peptides (tFT-$X_{12}$ library, replicate 1) with 100 permutations per peptide. The SHAP contribution scores for all features were then used to cluster the unstable peptides into 56 clusters using k-means[37,93]. The number of clusters was decided using the within sum of squares (WSS) method. In Fig. 2b and

Supplementary Fig. 2a, each cluster was depicted by a representative peptide, determined as the peptide sequence most similar to all other sequences in the cluster based on their local alignment using the Smith-Waterman algorithm[94]. Clusters with at least one mean SHAP sequence contribution score greater than 0.05 at positions −5 to −1 were considered to represent potential C-degron motifs (Supplementary Fig. 2). For these 21 clusters of potential C-degrons, logos were generated for sequence-based properties and position-specific biophysical properties, with positive values reflecting positive contributions to the unstable prediction. Mean contribution scores of global biophysical properties were displayed as bar graphs.

The clustering approach as a hypothesis generator was evaluated using the sets of Das1 and Doa10 degrons identified by MPS profiling of the tFT-X$_{12}$ degron library in the respective mutants (Fig. 5b, f). 759 Das1 degrons and 27 Doa10 degrons were in the set of 4110 putative degrons correctly classified by the model. The 21 putative C-degron clusters (Supplementary Fig. 2) contained 416 of the 759 Das1 degrons and only 3 of the 27 Doa10 degrons. Thus, the relative frequency of Das1 degrons is 2.2 fold higher in the putative C-degron clusters, whereas the relative frequency of Doa10 degrons is 6.7 fold higher in the other 35 clusters. This is consistent with the experimental classification of Das1 but not Doa10 degrons as C-degrons based on the C-terminal capping mutations (Figs. 4b, 5i, Supplementary Fig. 5d, Supplementary Fig. 6f).

### Colony fluorescence measurements

Yeast strains were assembled in 1536-colony format on agar plates using a pinning robot (Rotor, Singer Instruments). For each tFT strain, 4 replicates of the sample strain, 8 replicates of a control strain without a tFT and 4 replicates of a reference strain expressing a stable tFT construct, were arranged next to each other in 2 × 2 groups. Such ordered colony arrays were typically grown for 24 h at 30 °C before measuring colony fluorescence. Arrays with temperature-sensitive strains were first grown for 24 h at 30 °C, followed by 24 h of incubation at 37 °C before measuring colony fluorescence.

Fluorescence measurements were performed with a multimode microplate reader equipped with monochromators for precise selection of excitation and emission wavelengths (Spark, Tecan) and a custom temperature-controlled incubation chamber. Fluorescence intensities were measured as follows: mCherry with 586/10 nm excitation, 612/10 nm emission, optimal detector gain and 40 µs integration time; sfGFP with 488/10 nm excitation, 510/10 nm emission, optimal detector gain and 40 µs integration time. For each sample strain, fluorescence intensities were first corrected for background fluorescence by subtraction of the mean of neighboring non-fluorescent colonies, and then normalized by the mean of the neighboring reference colonies to correct for spatial effects. mCherry/sfGFP ratios were subsequently calculated for each sample replicate and summarized by the mean and standard deviation.

### Flow cytometry

Strains were grown overnight to saturation, diluted into fresh medium to 0.2 OD$_{600}$ and further grown for ~5 h to 0.8 OD$_{600}$. Single-cell fluorescence intensities were measured on a LSRFortessa SORP flow cytometer (BD Biosciences) using a 561 nm laser for mCherry excitation, a 600 nm long pass mirror and a 610/20 nm band pass filter for mCherry detection, a 488 nm laser for sfGFP excitation, a 505 nm long pass mirror and a 530/30 nm band pass filter for sfGFP detection (Supplementary Fig. 7e–g).

Measurements were gated for single cells with an sfGFP fluorescence intensity above the maximum intensity of a non-fluorescent control strain, and an mCherry/sfGFP ratio of each cell was calculated without background correction.

### Cycloheximide chases

Strains were grown to 0.8 OD$_{600}$, followed by addition of cycloheximide (C4859, Sigma-Aldrich) to a final concentration of 100 µg/ml to block protein synthesis. At each time point, ~8 × 10$^7$ cells were harvested by centrifugation. Whole cell extracts were prepared by alkaline lysis followed by precipitation with trichloroacetic acid[95] and separated by SDS-PAGE, followed by immunoblotting against GFP and against the loading control Pgk1 using mouse anti-GFP (1:2000 dilution, 11814460001, Roche), mouse anti-Pgk1 (1:1000 dilution, 459250, Thermo Fisher Scientific) and goat HRP-conjugated anti-mouse (1:5000 dilution, G-21040, Thermo Fisher Scientific) antibodies. Membranes were developed using a chemiluminescent horseradish peroxidase substrate (SuperSignal West Pico PLUS, 34580, Thermo Fisher Scientific) and imaged using a ChemiDoc MP imaging system (Bio-Rad). Quantification was performed with the associated ImageLab software, signals corresponding to a tFT fusion were normalized by the Pgk1 signal.

### Yeast two-hybrid

All yeast two-hybrid experiments were performed with the yKEK476 strain, which was derived from PJ69-4A[46]. yKEK476 lacks endogenous *DAS1* and carries a *HIS3* reporter. This strain was co-transformed with plasmids expressing Das1-AD and DBD-DHFR-peptide fusions. Transformants were selected on SC plates lacking leucine and tryptophan (SC-LW). Selected transformants were grown in SC-LW overnight to saturation and serial ten-fold dilutions were spotted on SC-LW plates as a control for growth or plates also lacking histidine (SC-LWH) to test for interactions between Das1-AD and DBD-DHFR-peptides. Growth on SC-LWH medium is only possible upon interaction between Das1-AD and DBD-DHFR-peptide constructs, which activates *HIS3* expression. Plates were photographed (Pheno-Booth imaging system, Singer Instruments) after 4 days of incubation at 30 °C.

### Targeted screens for factors involved in degradation of tFT-degron constructs

Query strains expressing tFT-X$_{12}$ constructs integrated into the *ura3* locus were crossed with an arrayed library of 138 strains carrying single deletions of most non-essential UPS components (Supplementary Table 1, Supplementary Data 2). Mating, diploid selection, sporulation, and haploid selection were performed in 1536-colony format following the SGA (synthetic genetic array) methodology[40,41] by sequential pinning of yeast colonies on agar plates with appropriate selective media using a pinning robot (Rotor, Singer Instruments). Each cross was performed in 4 technical replicates arranged next to each other in a 2 × 2 group, next to 8 replicates of a non-fluorescent control strain and 4 replicates of a reference strain expressing a stable tFT fusion.

The resulting haploid colony arrays carrying both a mutation in a UPS factor and a tFT or a control construct were grown for 24 h on SC agar plates lacking leucine. mCherry and sfGFP fluorescence intensities of colonies were measured with a multimode microplate reader (Spark, Tecan) as described above, but with two fixed detector gains to extend the detection range. In addition, the colony arrays were photographed both after the final haploid selection step and on the final measurement plates, to identify failed crosses based on colony size. After excluding measurements from failed crosses, fluorescence intensities were corrected for background fluorescence and spatial effects, as detailed above. mCherry/sfGFP intensity ratios were calculated per replicate. sfGFP intensities and mCherry/sfGFP ratios were subsequently normalized to the control cross with a wild type strain (*his3Δ::kanMX*) and log-transformed. Finally, technical replicates were summarized by the mean and standard deviation. A two-sided *t*-test was used to compare each sample cross to the wild type control cross and to compute *p*-values, adjusted for multiple testing using the method of Benjamini-Hochberg.

## Fluorescence microscopy

Strains were grown overnight to saturation, diluted into fresh medium to 0.2 OD$_{600}$ and further grown for ~5 h to 0.8 OD$_{600}$. Cells were collected by centrifugation of 1 ml of culture and resuspended in 10 μl of growth medium. Subsequently, 3 μl of the cell suspension were transferred onto a slide and immediately imaged on a confocal microscope (TCS SP5, Leica) using a 63x/1.4 NA oil immersion objective, a 488 nm laser at 10% power for sfGFP excitation and a 500–550 nm PMT detector with gain set to 650 to detect sfGFP emission.

## Statistics and reproducibility

In MPS profiling experiments, library complexity was ensured by analyzing a number of cells at least 50–100 fold larger than the number of sequence variants in a library. Two to three technical replicates corresponding to independent sorting and sequencing of the same yeast library were analyzed. MPS profiling of C-terminal capping and saturation mutagenesis libraries was performed once, as all variants of each peptide could be directly compared. SGA screens were performed in four technical replicates (four colonies of the same strain) placed next to each other on the screen plates. All other experiments were performed with at least two biological replicates, i.e., independent clones isolated from a single transformation, as is standard in the field.

## Reporting summary

Further information on research design is available in the Nature Portfolio Reporting Summary linked to this article.

## Data availability

The sequencing data generated in this study have been deposited in the Gene Expression Omnibus under accession code GSE246422. Processed MPS profiling data (peptide sequences and PSIs) are provided as Supplementary Data. The input data necessary for downstream analysis and data visualization are available at https://figshare.com/s/4c576dfd79e031878584. Source data are provided with this paper.

## Code availability

The pipeline for processing of MPS profiling data is available at https://github.com/Khmelinskii-Lab/Das1_C-degrons/tree/main/NGSpipe2go-MPSprofiling. The pipeline for downstream analysis and data visualization is available at https://github.com/Khmelinskii-Lab/Das1_C-degrons.

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

## Acknowledgements

We thank Michael Knop, Helle Ulrich, Elmar Schiebel and Zhaoyan Li for reagents, Katja Luck, Balca Mardin, Michael Knop and Helle Ulrich for critical reading of the manuscript. We thank the IMB Flow Cytometry Core Facility for the use of their instruments and cell sorting assistance. Funding of the German Research Foundation supported the BD FAC-SAria III SORP (P#210144599) and BD LSRFortessa SORP (P#210253511). Support by the IMB Genomics Core Facility and the use of its Next-Seq500 funded by the Deutsche Forschungsgemeinschaft (DFG, German Research Foundation – 329045328) is gratefully acknowledged. The IMB Media Lab and the IMB Protein Production facility are gratefully acknowledged for support with growth media and reagents. This work was supported by the European Research Council (ERC-2017-STG#759427 to A.K.).

## Author contributions

Conceptualization: K.Y.E.K. and A.K.; experimental investigation: K.Y.E.K.; analysis of genetic screens: K.Y.E.K.; analysis of MPS profiling experiments: K.Y.E.K., S.S. and F.R.; deep learning: S.S.; statistical analysis: S.S. and K.Y.E.K.; writing-original draft: A.K.; writing-review and editing: K.Y.E.K. and A.K.; supervision: K.Y.E.K. and A.K.; funding acquisition: A.K.

## Competing interests

The authors declare no competing interests.
