## [Peer Review File · Nature Communications]

REVIEWER COMMENTS

Reviewer #1 (Remarks to the Author):

Review of NCOMMS-23-34706

The manuscript by Kong et al. reports an impressive investigation of C-terminal degrons in yeast. A recent paper in Nature Communications, which similarly created a library of model proteins with varying C-terminal appendages, showed that many C-terminal degrons in yeast are defined by high hydrophobicity (Mashahreh et al.). Such hydrophobic degrons are also prominent in human cells (e.g., Koren et al.). It is noteworthy that the previous studies used longer C-terminal appendages than in Kong et al.—and this may influence the percentage of degrons recognized by various pathways. For example, the ubiquitin ligase Doa10 is thought to mainly target helical degrons (bearing hydrophobic amino acids), which could be relatively rare with only 12 amino acid appendages to a folded protein. Kong et al. focused mainly on degrons of 12 amino acids and of relatively low hydrophobicity, searching for sequence-specific degrons. In contrast to human cells, the existence of sequence-specific C-terminal degrons in yeast had not been obvious from previous studies. Using sophisticated computation, Kong et al. discovered that sequence-specific C-degrons also exist in yeast, albeit with different ‘rules’ and less specificity than in human cells. The manuscript also revealed a single ubiquitin ligase, SCFDas1, that recognizes a large percentage of the degrons tested in the study. SCFDas1 had been linked to degradation of a C-terminally extended Hac1 protein, but the broad specificity—and likely broad utility—of SCFDas1 was not appreciated from that single study (Di Santo et al., 2016). The manuscript from Kong et al. represents a major advance in our understanding of both SCFDas1 and C-terminal degrons. The manuscript is well written and the results are of high significance to the ubiquitin-proteasome field. We feel this manuscript is near ready for publication in Nature Communications. We have only a few comments that we would like to see addressed by the authors.

Main comments

1) Our main comment pertains to Figure 6 and corresponding text. Given that the full-length proteins studied are tagged (with the rather large dual fluorescent protein tag) and much of this figure still deals with model proteins only bearing the C-terminal 12 amino acids of yeast proteins, we feel that the manuscript doesn’t really show that “endogenous” proteins are affected by Das1. It would be nice to see one experiment with untagged proteins, even if overexpression is required to see the effect of *das1Δ*. We realize this may be difficult given the scarcity of antibodies for yeast proteins and the relatively small effects observed with Atg1 and Rpa12 (Fig. 6 k-l). Since we feel this important body of work should be published soon, we suggest at the least that “endogenous substrates” be changed to ‘yeast protein-derived substrates’. Perhaps in future work proteins genetically linked to DAS1 will provide a clear example of an endogenous substrate (Costanzo et al.). We noticed that the MSC1 gene has been linked to DAS1 in multiple studies. Interestingly, *Msc1* ends in asparagine (..FN).

2) Related to the above comment, page three of the manuscript introduces the ϕ N concept and lists I, L, M, V as “large hydrophobic amino acids”. Were these 4 amino acids chosen based on their properties (regardless of their emergence in degrons), or were they the ones that emerged from the degron analyses? Why are tryptophan and phenylalanine not included as large hydrophobic amino acids? F has a higher Kyte-Doolittle value than M. Is F less common than I, V, L, and M in the ϕ N degrons? Are the Das1 degrons best with large hydrophobic aliphatic amino acids?

Additional comments

1) Some of the figures are expansive, with small sub-figures. We suggest that any sub-figures that could be moved to supplemental be moved so that the main figures can be more concise. This could help make room for more detail needed in the main figure legends (see below). We were thinking Fig. 6g-j could be in supplemental, for example.

2) There is now an article in press at Journal of Biological Chemistry from the Taxis lab with similar findings/conclusions. This article should be cited and any relevant comparisons discussed.

3) Figure legends that could be improved:

1a: define tFT = tandem fluorescent protein timer

2: title: define ϕ (= ILMV)

2a: A short phrase describing the SHAP example in the right panel would be helpful.

3d: define SGA = synthetic genetic array

4: title: add “specificity for C-degrons”

6i,j: add number of technical replicates

4) Typo: discussion, third paragraph: should be “found”, not “finds”

Reviewer #2 (Remarks to the Author):

Reviewer #3 (Remarks to the Author):

Ka-Yiu Edwin Kong et al. present results of screening libraries of random peptides for C-degrons with the following data analysis. The manuscript is well written, all figures are readable and informative. I was able to access the supplementary codes deposited in the github repository.

- Literature review indicates that there have been previous attempts to use machine learning to predict protein stability index and data analysis. State of the art should be briefly presented in the Introduction section.
- Results section, paragraph "Identification of C-degron motifs". It is not clear if the presented performance of the model is for the training set, the test set, or combined. Please clarify.
- It would also be beneficial for the reader if the authors provide the performance of the ML model for the training and test sets separately (loss, accuracy, area under the curve (AUC), recall, precision, and mean squared error (MSE)). The values can be added to Figure S1d or presented as a separate table.
- The use of four test sets (instead of one) should be explained.
- Why were the PSI values taken from "replicate 1" for the construction of the data sets and not from the average of both replicates? From Figure S1a I can see that there is a high correlation between these values, but for the higher values of PSI the deviation is quite high.
- Using SHAP contribution scores for clustering is an interesting idea. However, the rationale behind such an analysis should be discussed.

Minor comments:

- the git repository address should be ``https://github.com/Khmelinskii-Lab/SCF-DAS1_substrates`` (without .git)

Reviewer #4 (Remarks to the Author):

The manuscript „ Orphan quality control by an SCF ubiquitin ligase directed to pervasive C-degrons“ by Kong et al deals with a very interesting topic, a protein quality control mechanism that ensures that orphaned complex subunits are degraded. The authors could implicate an SCF complex containing the F-box protein Das1 in this process in budding yeast. Moreover, the authors provide a comprehensive analysis of degradation-inducing sequences located at the C-terminus and characterize the degrons that are bound by Das1. Interestingly, about 40 % of the analyzed sequences were targeted by Das1 and relatively few by Doa10, an E3 that was implicated in protein quality control of C-terminal protein ends

before. Overall, the work is very interesting, however a few points need to be clarified before publication.

Major points:

1. The introduction is focused on N-degrons in yeast and C-degrons in mammalian cells. It lacks to mention previous work like Gilon et al Mol Cell Biol 1998 that should be added to the introduction and discussed in relation to the findings reported by the authors. The recent work of the groups of R. Hartmann-Petersen, K. Lindorf-Larsen, and T. Ravid on the topic of C-degrons is mentioned in passing and not discussed. Moreover, Hasenjäger et al JBC 2023 reported recently that a Das1-containing SCF complex is involved in a protein quality control pathway monitoring the C-terminus of proteins. Finally, the Koren lab reported recently (Makaros et al Mol Cell 2023) ubiquitin-independent proteasomal degradation driven by C-terminal degrons. The implementation of this literature and discussion of the reported results would significantly improve the manuscript.
2. The authors report the arbitrary selection of a PSI of 0.5 to distinguish between stable and unstable peptide-reporters. Such an important boundary should be backed up by experimental data. The authors should add correlative experiments between PSI and half life of the corresponding construct. Such a correlation should be given for several constructs within the whole PSI range.
3. The authors should improve the description of the constructs that were used to obtain the libraries. It seems that the authors did not add any unstructured linker between mCherry and the peptides that act as C-degron. As proteasomal degradation relies on an unstructured linker of about 30 amino acids to initiate substrate transfer into the inner proteasomal cavity for degradation, one wonders how relevant the data of the authors really is. The authors report that in case of the tFT-X4 and tFT-X8 libraries no degrons were present in the library. However, this might be due to insufficient sequence that could act as initiation sequence for proteasomal degradation. Moreover, in case of the tFT-X12 library the authors might have a bias towards sequences that destabilize the C-terminus of mCherry resulting in unstructured sequences that is long enough to act as initiation sequence. In a recent publication by Hasenjäger et al, degron sequences were reported by random exchanges of the last two, three or six residues. The authors need to clearly show that the tFT-X12 library does not contain a bias towards sequences destabilizing the end of mCherry and the nature of the absence of degron sequences in case of the X4 and X8 libraries. This problem does not affect the results concerning Das1 as an E3 that targets C-degrons, but the statement about enrichment of hydrophobic amino acids in Das1-dependent C-degrons might be affected by a potential bias in the tFT-X12 sequences as well as the claim to provide a comprehensive view on the C-degronome in yeast.
4. Figure 2f: The authors used libraries with specific amino acids at fixed positions to show that hydrophobic amino acids at -2 and N at -1 are decisive degron constituents. However, only about 10 % of the peptides were destabilizing, 90 % of the peptides were stable. Which features in the 10 amino acids upstream of the last 2 residues leads to a degron? Are their further degron constituents present there? The differentiation between stable and destabilized peptide with fixed 2 amino acids at the end might also be influenced by the potential bias described above.
5. Fig 3c: How do the authors explain the discrepancy between fluorescence data and the chx chase experiments? Are 20 % of the identified degrons actually not degradation inducing sequences? Which

PSI do the excluded sequences LN2, MN1 and MN2? The authors should measure half-lives for all the degnon constructs shown in Fig3c and control constructs to clarify this point.

6. Page 4, end of 4th paragraph: the authors neglect here the literature about doa10-dependent C-terminal degnons. See also comment 1.

7. How were the technical replicates performed that are mentioned in the figure legends. This is not clearly explained and is especially important for western blotting, in which almost each step of the procedure could be technically replicated. For most measurements, the usage of biological replicates would be an improvement.

Minor points

8. Page 4 paragraph 5: the authors seem to use a very strict definition of a C-degnon: "...Doa10-dependent turnover of tFT-tagged R1 to R4 peptides, indicating that these are not C-degnons". Such a strict definition that a C-degnon has to include the last amino acid is not used in the current literature about C-degnons and is also not used as strictly for N-degnons (see GID-dependent proline containing degnons with proline at position 2). Rephrase?

9. Fig 6i to l: Which half-lives have the constructs with and without overexpression shown in this figure? The RFP/GFP ratios that are shown are in the same range as the RFP/GFP ratios of constructs defined as stable (tFT-X12S Figure S4, tFT-X12S Figure 3). please explain.

10. Discussion: "Das1 appears to have an exceptionally broad specificity,...". Is Das1 so exceptional in its broad specificity? Ubr1 has 3 binding sites for degnon binding, Doa10 recognizes degnons in membrane proteins at the cytosolic face of the ER membrane, inside the ER membrane as well as degnons in cytosolic substrates. Maybe this emphasis of an exceptionally broad specificity of Das1 is exaggerated, as other well studied E3s in yeast show similarly a broad specificity. The explanation that follows after the initial statement of Das1 as exceptional could also be used to describe Ubr1-dependent N-degnons.

We thank all reviewers for their constructive criticism and suggestions. We have addressed all comments as detailed below.

Reviewer #1

The manuscript by Kong et al. reports an impressive investigation of C-terminal degrons in yeast. A recent paper in Nature Communications, which similarly created a library of model proteins with varying C-terminal appendages, showed that many C-terminal degrons in yeast are defined by high hydrophobicity (Mashahreh et al.). Such hydrophobic degrons are also prominent in human cells (e.g., Koren et al.). It is noteworthy that the previous studies used longer C-terminal appendages than in Kong et al.—and this may influence the percentage of degrons recognized by various pathways. For example, the ubiquitin ligase Doa10 is thought to mainly target helical degrons (bearing hydrophobic amino acids), which could be relatively rare with only 12 amino acid appendages to a folded protein. Kong et al. focused mainly on degrons of 12 amino acids and of relatively low hydrophobicity, searching for sequence-specific degrons. In contrast to human cells, the existence of sequence-specific C-terminal degrons in yeast had not been obvious from previous studies. Using sophisticated computation, Kong et al. discovered that sequence-specific C-degrons also exist in yeast, albeit with different ‘rules’ and less specificity than in human cells. The manuscript also revealed a single ubiquitin ligase, SCFDas1, that recognizes a large percentage of the degrons tested in the study. SCFDas1 had been linked to degradation of a C-terminally extended Hac1 protein, but the broad specificity—and likely broad utility—of SCFDas1 was not appreciated from that single study (Di Santo et al., 2016). The manuscript from Kong et al. represents a major advance in our understanding of both SCFDas1 and C-terminal degrons. The manuscript is well written and the results are of high significance to the ubiquitin-proteasome field. We feel this manuscript is near ready for publication in Nature Communications. We have only a few comments that we would like to see addressed by the authors.

Main comments

1) Our main comment pertains to Figure 6 and corresponding text. Given that the full-length proteins studied are tagged (with the rather large dual fluorescent protein tag) and much of this figure still deals with model proteins only bearing the C-terminal 12 amino acids of yeast proteins, we feel that the manuscript doesn’t really show that “endogenous” proteins are affected by Das1. It would be nice to see one experiment with untagged proteins, even if overexpression is required to see the effect of *das1*Δ. We realize this may be difficult given the scarcity of antibodies for yeast proteins and the relatively small effects observed with Atg1 and Rpa12 (Fig. 6 k-l). Since we feel this important body of work should be published soon, we suggest at the least that “endogenous substrates” be changed to ‘yeast protein-derived substrates’. Perhaps in future work proteins genetically linked to DAS1 will provide a clear example of an endogenous substrate (Costanzo et al.). We noticed that the MSC1 gene has been linked to DAS1 in multiple studies. Interestingly, Msc1 ends in asparagine (..FN).

We replaced the term “endogenous substrates” throughout the manuscript with “full-length yeast proteins” or “full-length substrates”.

2) Related to the above comment, page three of the manuscript introduces the ϕ N concept and lists I, L, M, V as “large hydrophobic amino acids”. Were these 4 amino acids chosen based on their properties (regardless of their emergence in degrons), or were they the ones that emerged from the degron analyses? Why are tryptophan and phenylalanine not included as large hydrophobic amino acids? F has a higher Kyte-Doolittle value than M. Is F less common than I, V, L, and M in the ϕ N degrons? Are the Das1 degrons best with large hydrophobic aliphatic amino acids?

The four amino acids ILMV emerged from the analysis of degrons in the tFT-X₁₂ library. In clusters 31 and 45 in particular, ILV at position -2 were the 3 hydrophobic amino acids with largest SHAP contribution scores (Fig. 2c). In addition, IN, LN and VN (and to some extent MN), but not FN or WN, were among the most destabilizing dipeptides when located at the C-terminus compared to internal positions (Fig. 2e). We now clarified this in the Results section corresponding to Fig. 2c.

This initial hypothesis is in line with our saturation mutagenesis experiments in Fig. 4f and Fig. S6g, which show that [ILMV] are preferred, whereas [FW] are allowed but suboptimal, amino acids at position -2. This is now stated in the Results section corresponding to Fig. 4f.

Additional comments

1) Some of the figures are expansive, with small sub-figures. We suggest that any sub-figures that could be moved to supplemental be moved so that the main figures can be more concise. This could help make room for more detail needed in the main figure legends (see below). We were thinking Fig. 6g-j could be in supplemental, for example.

We simplified Fig. 3. Validation of the SGA screens with fluorescence measurements and immunoblots of the cycloheximide chases in *das1Δ* and SCF mutants are now in Fig. S4b, e.

We simplified Fig. 6. C-terminal capping and saturation mutagenesis of Atg1 and Rpa12 C-degrons are now in Fig. S6f, g.

2) There is now an article in press at Journal of Biological Chemistry from the Taxis lab with similar findings/conclusions. This article should be cited and any relevant comparisons discussed.

We now reference the Hasenjäger et al. study in the discussion as follows:

“It is striking that ~36% of random degrons (1678 out of 4709) and ~54% of degrons in the yeast C-terminome (246 out of 458) are C-degrons targeted by SCF^{Das1}, arguing that the SCF/C-degron pathway has a broad specificity towards C-degrons. Further supporting this notion, Hasenjäger et al. independently identified SCF^{Das1} as a factor targeting a wide range of C-degrons⁶³.”

20 of the 25 Das1 degrons identified by Hasenjäger et al. have at least one of the determinants identified in our experiments (W at -5, K at -3, [ILM] at -2 or [CN] at -1). However, this set of Das1 degrons is biased as 15 of them originate from a semi-random peptide library with a fixed K at position -3 (the AEA**K**XX library), making such a comparison of dubious value.

3) Figure legends that could be improved:

1a: define tFT = tandem fluorescent protein timer

2: title: define ϕ (= ILMV)

2a: A short phrase describing the SHAP example in the right panel would be helpful.

3d: define SGA = synthetic genetic array

4: title: add “specificity for C-degrons”

6i,j: add number of technical replicates

We modified the figure legends following these suggestions.

4) Typo: discussion, third paragraph: should be “found”, not “finds”

Corrected.

Reviewer #2

Reviewer #3

Ka-Yiu Edwin Kong et al. present results of screening libraries of random peptides for C-degrons with the following data analysis. The manuscript is well written, all figures are readable and informative. I was able to access the supplementary codes deposited in the github repository.

- Literature review indicates that there have been previous attempts to use machine learning to predict protein stability index and data analysis. State of the art should be briefly presented in the Introduction section.

We now reference prior work on predicting degrons as follows:

“Despite the prevalence of hydrophobic degrons, we sought to identify sequence-specific degrons and degron motifs in our dataset. Previous efforts to predict degrons have used a variety of approaches, including deep learning^{33–36}. Here we decided to combine a deep learning approach with model interpretation using SHapley Additive exPlanations (SHAP)³⁷.”

- Results section, paragraph "Identification of C-degron motifs". It is not clear if the presented performance of the model is for the training set, the test set, or combined. Please clarify.

We clarified the statement as follows: “After 30 training epochs, the model achieved an overall training accuracy of 0.85 (Fig. S1f, Methods)”.

- It would also be beneficial for the reader if the authors provide the performance of the ML model for the training and test sets separately (loss, accuracy, area under the curve (AUC), recall, precision, and mean squared error (MSE)). The values can be added to Figure S1d or presented as a separate table.

We added the values for the model performance for the training and test sets to Fig. S1f.

- The use of four test sets (instead of one) should be explained.

Four test sets were used: test sets 1-3 had no peptides in common whereas test set 4 was formed by combining test sets 1-3. The four test sets were used to ensure that assessment of the model performance is not affected by the selection of the test set.

This information is now included in the Methods section.

- Why were the PSI values taken from "replicate 1" for the construction of the data sets and not from the average of both replicates? From Figure S1a I can see that there is a high correlation between these values, but for the higher values of PSI the deviation is quite high.

Experiments with the degron library in Fig. 5 were performed based on the analysis of replicate 1 of the tFT-X₁₂ library. Replicate 2 of the tFT-X₁₂ library was performed and analyzed at a later stage. For this reason, a model trained on replicate 1 alone was used for consistency between the analyses shown in Fig.1 and Fig.5.

Nevertheless, we tested the performance of the model (trained on replicate 1) on replicate 2, and on the average of replicates 1 and 2.

47876 peptides with at least 10 sequencing reads were detected in replicate 2, 4837 unstable (PSI < 0.5) and 43039 stable peptides. The model correctly classified 4212 unstable and 38396 stable peptides.

Replicates 1 and 2 were combined using linear regression. Out of 43095 peptides in the combined dataset, 3973 of the 4412 unstable (mean PSI < 0.5) and 34591 of the 38683 stable peptides were correctly classified by the model.

The performance metrics were similar to those seen with replicate 1.

	loss	accuracy	AUC	precision	recall	MSE
replicate 1 (test set 4)	0.36	0.82	0.91	0.82	0.82	0.11
replicate 2	0.30	0.89	0.94	0.89	0.89	0.09
combined	0.29	0.89	0.95	0.89	0.89	0.09

This information is now included in the Methods section.

- Using SHAP contribution scores for clustering is an interesting idea. However, the rationale behind such an analysis should be discussed.

We expanded our explanation of SHAP and the rationale for clustering in the Results section as follows:

“We trained a deep neural network model to classify peptides into unstable (putative degrons) or stable groups (Fig. 2a, Methods). The model takes as features the sequence of a peptide and six of its biophysical properties, which correlated with PSI in the tFT-X₁₂ library (Supplementary Fig. 1e), so as to identify sequence-specific degrons for which overall hydrophobicity is not an important determinant. After 30 training epochs, the model achieved an overall training accuracy of 0.85 (Supplementary Fig. 1f, Methods), correctly classified 4110 of the 4726 peptides with PSI < 0.5 in the tFT-X₁₂ library as unstable and 37330 of the 41426 peptides with PSI > 0.5 as stable. Next, we used SHAP to interpret the model predictions. The goal of SHAP is to explain the classification of a given peptide as stable or unstable by computing the contribution of each feature (amino acid at each position, biophysical properties per position and of the peptide as a whole) to the prediction, in the form of SHAP values or contribution scores, and in this way understand features of degrons (Fig. 2a, Methods). SHAP contribution scores for the 4110 correctly classified putative degrons showed that the peptide sequence contributed the most to the prediction (Supplementary Fig. 1g). Both positional and global biophysical properties, the aliphatic index in particular, also contributed to the classification.

To identify potential degron motifs, we clustered the 4110 putative degrons based on the vectors of SHAP contribution scores (Methods). Although SHAP contribution scores can be interpreted for each individual peptide (Fig. 2a), we reasoned that clustering would identify similar peptides and define common features learned by the model that result in their classification as putative degrons, leading to putative motifs. This approach yielded 56 clusters, including 21 clusters with mean sequence contribution scores suggestive of C-degron motifs (at least one mean sequence contribution score above 0.05 at positions -5 to -1, Methods), containing a total of 1476 peptides (Fig. S2a, b).”

In addition, we evaluated the clustering approach as a hypothesis generator using the sets of Das1 and Doa10 degrons identified by MPS profiling of the tFT-X₁₂ degron library in the respective mutants (Fig. 5). 759 Das1 degrons and 27 Doa10 degrons were in the set of 4110 putative degrons correctly classified by the model. The 21 putative C-degron clusters (Fig. S2a, b) contained 416 of the 759 Das1 degrons and only 3 of the 27 Doa10 degrons. Thus, the relative frequency of Das1 degrons is 2.2 fold higher in the putative C-degron clusters, whereas the relative frequency of Doa10 degrons is 6.7 fold higher in the other 35 clusters. This is consistent with the experimental classification of Das1 but not Doa10 degrons as C-degrons based on the C-terminal capping mutations (Fig. 4b, 5i, S5d, S6f).

This information is now included in the Methods section.

Minor comments:

- the git repository address should be `https://github.com/Khmelinskii-Lab/SCF-DAS1_substrates` (without .git)

Corrected.

Reviewer #4

The manuscript „Orphan quality control by an SCF ubiquitin ligase directed to pervasive C-degrons“ by Kong et al deals with a very interesting topic, a protein quality control mechanism that ensures that orphaned complex subunits are degraded. The authors could implicate an SCF complex containing the F-box protein Das1 in this process in budding yeast. Moreover, the authors provide a comprehensive analysis of degradation-inducing sequences located at the C-terminus and characterize the degrons that are bound by Das1. Interestingly, about 40 % of the analyzed sequences were targeted by Das1 and relatively few by Doa10, an E3 that was implicated in protein quality control of C-terminal protein ends before. Overall, the work is very interesting, however a few points need to be clarified before publication.

Major points:

1. The introduction is focused on N-degrons in yeast and C-degrons in mammalian cells. It lacks to mention previous work like Gilon et al Mol Cell Biol 1998 that should be added to the introduction and discussed in relation to the findings reported by the authors. The recent work of the groups of R. Hartmann-Petersen, K. Lindorf-Larsen, and T. Ravid on the topic of C-degrons is mentioned in passing and not discussed. Moreover, Hasenjäger et al JBC 2023 reported recently that a Das1-containing SCF complex is involved in a protein quality control pathway monitoring the C-terminus of proteins. Finally, the Koren lab reported recently (Makaros et al Mol Cell 2023) ubiquitin-independent proteasomal degradation driven by C-terminal degrons. The implementation of this literature and discussion of the reported results would significantly improve the manuscript.

We now refer to the Makaros et al. study in the Discussion as follows:

“Our survey of C-terminal degrons using libraries of 12 amino acid long peptides revealed that ~10% of random peptides (4726 out of 46152 sequences in the tFT-X₁₂ library) and ~7% of yeast C-termini (458 out of 6403 sequences in the yeast C-terminome library) encode degrons. It is possible that additional degrons could be identified using reporters including unstructured regions of different lengths and sequence composition as initiation sites of proteasomal degradation^{28,29,61}. The occurrence of ubiquitin-independent degrons in these libraries, as recently uncovered in human cells⁶², remains to be determined.”

We now reference the Hasenjäger et al. study in the Discussion as follows:

“It is striking that ~36% of random degrons (1678 out of 4709) and ~54% of degrons in the yeast C-terminome (246 out of 458) are C-degrons targeted by SCF^{Das1}, arguing that the SCF/C-degron pathway has a broad specificity towards C-degrons. Further supporting this notion, Hasenjäger et al. independently identified SCF^{Das1} as a factor targeting a wide range of C-degrons⁶³.”

Please see also our response to reviewer 1, additional comment 2.

We now reference Gilon et al. 1998 (PMID: 9582269), Ravid et al. 2006 (PMID: 16437165), Maurer et al. 2016 (PMID: 27172186) and Johansson et al. 2023 (PMID: 36495918). To the best of our knowledge, there is no evidence that Doa10 recognizes C-degrons. (Please see also our responses to point 6 on Doa10 literature and to point 8 on the definition of C-degrons.)

Doa10 is included in our analysis as an E3 targeting a variety of hydrophobic degrons for comparison with Das1 and sequence-specific C-degrons. For these reasons, the discussion of the Doa10 findings is largely confined to the Results section.

2. The authors report the arbitrary selection of a PSI of 0.5 to distinguish between stable and unstable peptide-reporters. Such an important boundary should be backed up by experimental data. The authors should add correlative experiments between PSI and half life of the corresponding construct. Such a correlation should be given for several constructs within the whole PSI range.

The 0.5 PSI threshold is based on the experimental distribution of PSIs in the tFT-X₁₂ library.

The distribution of PSIs in the tFT-X₁₂ library is unimodal, centered at 0.668 ± 0.046 (median \pm median absolute deviation (mad)) and skewed towards low PSIs. Most constructs in the library are likely stable as the distribution of mCherry/sfGFP ratios in the library is close to a strain expressing only the tFT (Fig. 1b).

A dispersion $d = 3.7 \cdot \text{mad}$ was chosen based on this distribution of PSIs. ~4-5% of the PSIs are expected to be outside of the median $\pm d$ range for a symmetric distribution. For the experimental distribution, 2% of PSIs are above the median + d (0.84) threshold and 10% of PSI are below the median - d (0.5) threshold.

This information is now included in the Methods section and provided in Fig. S1d.

(Top) distribution of PSIs in the tFT-X₁₂ library (replicate 1). The median $\pm 3.7 \cdot \text{mad}$ range is highlighted in red. The percentage of PSIs outside of this range is indicated.

(Bottom) correlation between peptide hydrophobicity and PSI in the tFT-X₁₂ library from Fig. 1e.

Based on the relationship between PSI and half-lives measured in CHX chases for the 11 constructs in Fig. 3a, c, we estimate that the 0.5 PSI threshold corresponds to a half-life of ~260 min. As detailed below in response to point 5, this is at the limit of what we can reliably validate and distinguish from a stable construct with CHX chases.

Correlation between PSI (profiling of the tFT-X₁₂ library, Fig. 1e) and half-life (determined with CHX chases, Fig. 3c) for the indicated tFT-X₁₂ constructs listed in Fig. 3a.

A PSI of 0.5 corresponds to a half-life of 262 min (dashed lines) based on the linear fit in red.

This information is now provided in Fig. S3g.

3. The authors should improve the description of the constructs that were used to obtain the libraries. It seems that the authors did not add any unstructured linker between mCherry and the peptides that act as C-degron. As proteasomal degradation relies on an unstructured linker of about 30 amino acids to initiate substrate transfer into the inner proteasomal cavity for degradation, one wonders how relevant the data of the authors really is. The authors report that in case of the tFT-X₄ and tFT-X₈ libraries no degrons were present in the library. However, this might be due to insufficient sequence that could act as initiation sequence for proteasomal degradation. Moreover, in case of the tFT-X₁₂ library the authors might have a bias towards sequences that destabilize the C-terminus of mCherry resulting in unstructured sequences that is long enough to act as initiation sequence. In a recent publication by Hasenjäger et al, degron sequences were reported by random exchanges of the last two, three or six residues. The authors need to clearly show that the tFT-X₁₂ library does not contain a bias towards sequences destabilizing the end of mCherry and the nature of the absence of degron sequences in case of the X₄ and X₈ libraries. This problem does not affect the results concerning Das1 as an E3 that targets C-degrons, but the statement about enrichment of hydrophobic amino acids in Das1-dependent C-degrons might be affected by a potential bias in the tFT-X₁₂ sequences as well as the claim to provide a comprehensive view on the C-degronome in yeast.

The C-terminus of mCherry (RAEGRHSTGGMDELYK) is separated from the assayed peptides by the short GGG linker in our constructs. This information is now included in the Methods section.

Two factors could potentially contribute to the apparent lack of degrons in the tFT-X₄ and tFT-X₈ libraries: the relative inaccessibility of C-terminal peptides due to the short linker separating them from mCherry and the lack of a long unstructured region that could serve as an initiation site for proteasomal degradation. This statement is now included in the Results section.

Our results also show that a 30 amino acid long unstructured region is not absolutely required for efficient proteasomal degradation *in vivo*. For instance, over 1000 peptides act as Das1 degrons in the context of our reporter, despite the lack of said long unstructured region. Nevertheless, it is possible that additional degrons could be identified using reporters with unstructured regions of different lengths and sequence composition. This caveat is now included in the Discussion and the claim of a comprehensive view on the yeast C-degronome was removed from the manuscript.

On destabilization of the mCherry C-terminus by the fused peptides. Because *Aequorea victoria* green fluorescent protein (GFP) is relatively indifferent to N- or C-terminal fusions, the N- and C- termini of GFP were used in the design and evolution of commonly used fluorescent proteins (FPs) to make them tolerant to N- and C-terminal fusions. This includes mCherry, which starts with the first seven amino acids (MVSKGEE) and ends with the last seven amino acids of eGFP (GMDELYK) (Shaner et al. 2004 PMID: 15558047), mNeonGreen, mScarlet and others. Moreover, destabilization of mCherry by C-terminal fusions would likely lead to loss of fluorescence as the last beta strand in GFP and mCherry folds is important for fluorophore maturation, a feature exploited in the split GFP1-10/GFP11 system (Cabantous et al. 2005 PMID: 15580262, Kamiyama et al. 2016 PMID: 26988139). Although such an effect cannot be excluded, it is likely a rare occurrence as estimates of protein abundance across the proteome using different approaches (fluorescence measurements

of strains expressing ORFs tagging with various FPs including mCherry, CHX chases of strains with TAP-tagged ORFs or mass spectrometry of untagged strains) are generally in good agreement (Ho et al. 2018 PMID: 29361465, Weill et al. 2018 PMID: 29988094, Meurer et al. 2018 PMID: 29988096).

4. Figure 2f: The authors used libraries with specific amino acids at fixed positions to show that hydrophobic amino acids at -2 and N at -1 are decisive degron constituents. However, only about 10 % of the peptides were destabilizing, 90 % of the peptides were stable. Which features in the 10 amino acids upstream of the last 2 residues leads to a degron? Are their further degron constituents present there? The differentiation between stable and destabilized peptide with fixed 2 amino acids at the end might also be influenced by the potential bias described above.

As we show in Fig. 4 with saturation mutagenesis of IN2, LN1, LN3 and VN2 degrons, there are preferred amino acids also at positions -5 to -3, whereby negatively charged amino acids, glycine, proline and serine are generally disfavored. Please see also our response to point 2 of reviewer 1.

Fig. 2f refers to the percentage of cells with unstable variants (percentage of cells in the pool with an mCherry/sfGFP ratio below a threshold), not percentage of degrons in each library. The percentage of cells with low mCherry/sfGFP ratios is a reasonable proxy for the frequency of degrons and can be used to compare different libraries (as done for example by Yeh et al. 2021 PMID: 33469951). But it cannot be used as a direct measure of degron frequency, as distributions of sequence variants in pooled libraries are typically log normal and can span 1-3 orders of magnitude.

In addition, the mCherry/sfGFP threshold used in Fig. 2f and Fig. S3a, b is more stringent than the PSI threshold of 0.5. These two factors explain the difference between the percentage of degrons in the tFT-X₁₂ library determined by MPS profiling (~10%, Fig. 1c) and the percentage of cells with unstable variants in the same library estimated by flow cytometry (~3.6%, Fig. 1b, Fig. 2f). To clarify this, we now include this information in the legend to Fig. S3a, b.

5. Fig 3c: How do the authors explain the discrepancy between fluorescence data and the chx chase experiments? Are 20 % of the identified degrons actually not degradation inducing sequences? Which PSI do the excluded sequences LN2, MN1 and MN2? The authors should measure half-lives for all the degron constructs shown in Fig3c and control constructs to clarify this point.

The LN2, MN1 and MN2 sequences are Das1 degrons. Please see our response to point 2 for the correlation between PSI and half-life for the constructs in Fig. 3c.

We observed that the tFT-LN2, tFT-MN1 and tFT-MN2 fusions were fully stabilized in the absence of *DAS1* when profiling the tFT-X₁₂ degron library (Fig. 5, Table S5). This is now shown in Fig. S5h.

PSIs of the indicated constructs determined by MPS profiling of the tFT-X₁₂ degron library in the wild type, *das1*Δ and *doa10*Δ backgrounds at 30°C.

A stable control peptide, s1 (ESCWVSRVGVCR), is shown for comparison (mean ± s.d., n = 2 (*doa10*Δ) or 3).

The LN2, MN1 and MN2 constructs were not used in the SGA screens that led to the identification of Das1 because their turnover is at the limit of what we can reliably validate with the independent approach used in our manuscript – CHX chases.

Compared to measurements of protein half-lives with cycloheximide chases (or pulse-chase experiments), the tFT assay has higher precision and broader dynamic range. Protein half-lives between ~10 min and ~8 h can be distinguished with the mCherry-sfGFP timer (Supplementary Fig. 8b,c in Khmelinskii et al. 2012 PMID: 22729030). For comparison, treatment of yeast cells with CHX for more than 2 h leads to significant depletion of the free ubiquitin pool, especially affecting estimates of half-lives for proteins with slower turnover. This relatively short 2 h CHX treatment period also limits the range of half-lives that can be reliably measured in this assay. For instance, to precisely measure a protein half-life of ~8 h would require reliably detecting a change in protein abundance of 12.5% in the 2 h chase period.

Thus, although the tFT-LN2, tFT-MN1 and tFT-MN2 constructs are unstable in the profiling of the tFT-X₁₂ library (PSIs of 0.31, 0.37 and 0.32, respectively compared to 0.67 for the tFT-X₁₂S construct), unstable in the flow cytometry tFT assay (mCherry/sfGFP ratios of 0.40±0.03, 0.46±0.02 and 0.41±0.03, respectively, compared to 1.00±0.02 for the tFT-X₁₂S construct) and exhibit measurable turnover in the CHX chase assay (half-lives of ~130, 200 and 110 min compared to 410 min for the tFT-X₁₂S construct), these constructs were not used in the SGA screens.

We have clarified this in the Results section.

6. Page 4, end of 4th paragraph: the authors neglect here the literature about doa10-dependent C-terminal degrons. See also comment 1.

To the best of our knowledge, there is no evidence that Doa10 recognizes C-degrons. (Please see our response to point 1 about additional literature on Doa10 degrons and to point 8 on the definition of C-degrons.)

For example, the CL1 degron, identified by Gilon et al. 1998 and used in a variety of model systems, does not have to be located at the extreme C-terminus to drive Doa10-dependent turnover of a reporter. Below we show colony fluorescence measurements of strains expressing tFT-CL1 constructs, either wild type, with an additional W at the C-terminus (+1W) or fused to the X₁₂S peptide (+X₁₂S) at the C-terminus.

7. How were the technical replicates performed that are mentioned in the figure legends. This is not clearly explained and is especially important for western blotting, in which almost each step of the procedure could be technically replicated. For most measurements, the usage of biological replicates would be an improvement.

In MPS profiling, technical replicates corresponding to independent sorting and sequencing of the same yeast library were analyzed. SGA screens were performed in four technical replicates (four colonies of the same strain) placed next to each other on the screen plates. All other experiments, including CHX chases, were performed with biological replicates, i.e., independent clones isolated from a single transformation.

We consolidated this information in one paragraph in the Methods section.

Minor points

8. Page 4 paragraph 5: the authors seem to use a very strict definition of a C-degron: "...Doa10-dependent turnover of tFT-tagged R1 to R4 peptides, indicating that these are not C-degrons". Such a strict definition that a C-degron has to include the last amino acid is not used in the current literature about C-degrons and is also not used as strictly for N-degrons (see GID-dependent proline containing degrons with proline at position 2). Rephrase?

As defined by Varshavsky in a 2019 perspective (PMID: 30622213), “N-degrons and C-degrons are degradation signals whose main determinants are, respectively, the N-terminal and C-terminal residues of cellular proteins”. Timms and Koren use essentially the same definition in their 2020 review (PMID: 32627813), “(...) an analogous set of C-degron pathways operate in parallel on a suite of degron motifs located at the extreme C-termini of proteins”.

This definition is consistently used throughout the works of the Varshavsky, Elledge, Yen, Koren, Knop and other laboratories, starting with the pioneering discovery of the N-end rule pathway in 1986, and is followed in our manuscript.

The GID complex does not recognize degrons with proline at position 2. The N-termini of Gid4 substrates with proline at position 2, Pck1 (SPSKMN) and Aro10 (APVTIE), are pro-N-degrons that must be processed by aminopeptidases to expose the N-degrons starting with proline (Chen et al. 2021 PMID: 34663735).

9. Fig 6i to l: Which half-lives have the constructs with and without overexpression shown in this figure? The RFP/GFP ratios that are shown are in the same range as the RFP/GFP ratios of constructs defined as stable (tFT-X12S Figure S4, tFT-X12S Figure 3). please explain.

The measurements previously shown in Fig. 6i-l were not normalized to the stable reference construct tFT-X₁₂S. To correct this, we repeated the experiments, including the tFT-X₁₂S construct for comparison. It is now clear that in wild-type cells, the tFT-Atg1⁻¹²⁻¹, tFT-Rpa12⁻¹²⁻¹, tFT-Atg1 and tFT-Rpa12 constructs are significantly less stable than tFT-X₁₂S. This data replaces the previous analysis and is now shown in Fig. 6g-j.

10. Discussion: “ Das1 appears to have an exceptionally broad specificity,...”. Is Das1 so exceptional in its broad specificity? Ubr1 has 3 binding sites for degron binding, Doa10 recognizes degrons in membrane proteins at the cytosolic face of the ER membrane, inside the ER membrane as well as degrons in cytosolic substrates. Maybe this emphasis of an exceptionally broad specificity of Das1 is exaggerated, as other well studied E3s in yeast show similarly a broad specificity. The explanation that follows after the initial statement of Das1 as exceptional could also be used to describe Ubr1-dependent N-degrons.

In our statement, we compared Das1 to other E3s targeting specifically C-degrons. Nevertheless, the comparison to Ubr1 is interesting.

The broad specificity of Das1 towards C-degrons is indeed reminiscent of the broad specificity of Ubr1 towards N-degrons. However, Ubr1 is a large ubiquitin ligase (225 kDa) that recognizes a variety of N-degrons and internal degrons using three distinct substrate-binding sites. It will be interesting to determine whether one or multiple substrate-binding sites in the substantially smaller Das1 (77 kDa) are responsible for its broad specificity.

This information is now included in the Discussion section.

REVIEWERS' COMMENTS

Reviewer #4 (Remarks to the Author):

My concerns have been adequately addressed. I recommend publication.